# Shifting the Breaking Point of Flow Matching for Multi-Instance Editing

**Carmine Zaccagnino** [* 1]   **Fabio Quattrini** [* 1]   **Enis Simsar** [2]   **Marta Tintoré Gazulla** [3]   **Rita Cucchiara** [1]
**Alessio Tonioni** [3]   **Silvia Cascianelli** [1]

## Abstract

Flow matching models have recently emerged as an efficient alternative to diffusion, especially for text-guided image generation and editing, offering faster inference through continuous-time dynamics. However, existing flow-based editors predominantly support global or single-instruction edits and struggle with multi-instance scenarios, where multiple parts of a reference input must be edited independently without semantic interference. We identify this limitation as a consequence of globally conditioned velocity fields and joint attention mechanisms, which entangle concurrent edits. To address this issue, we introduce Instance-Disentangled Attention, a mechanism that partitions joint attention operations, enforcing binding between instance-specific textual instructions and spatial regions during velocity field estimation. We evaluate our approach on both natural image editing and a newly introduced benchmark of text-dense infographics with region-level editing instructions. Experimental results demonstrate that our approach promotes edit disentanglement and locality while preserving global output coherence, enabling single-pass, instance-level editing.

## 1. Introduction

High-fidelity text-guided image editing has traditionally relied on diffusion models with U-Net architectures (Sohl-Dickstein et al., 2015; Ho et al., 2020; Song et al., 2021) for global and localized edits. These models iteratively denoise a sample in latent or pixel space, enabling a wide range of operations such as attribute modification, object insertion or removal, and inpainting. In particular, given an input image and a text condition, editing capabilities are achieved with techniques such as masked denoising, cross-attention manipulation, and inversion, enabling object-level control and compositional editing (Hertz et al., 2023; Mokady et al., 2023; Avrahami et al., 2022; Goel et al., 2024; Couairon et al., 2023). Despite these successes, most works and benchmarks focus on single or at most few-instance edits in the natural image domain (Yang et al., 2024; Matsuda et al., 2024; Fu et al., 2026; Chakrabarty et al., 2024), offering limited support for *multi-instance editing*. This task requires the model to yield *edits disentanglement*, *i.e.*, multiple edits should be independently controllable and combinable, *edits locality*, *i.e.*, non-edited regions should remain unchanged, and *global coherence*, *i.e.*, the whole image should still be visually coherent. Moreover, when dealing with an increasingly high number of edits $N$, computational constraints of some methods increasingly bottleneck inference pipelines.

Recently, the image generation community is shifting towards Multimodal Diffusion Transformers (MMDiTs) (Esser et al., 2024) trained with Rectified Flow Matching (Lipman et al., 2023). Compared to diffusing stochastic processes, this framework offers higher visual quality and faster inference by modeling generation as an Ordinary Differential Equation (ODE) and learning to predict velocity fields $v_\theta$ over straight paths between data and noise distribution. The MMDiT architecture iteratively refines image tokens from the random noise distribution into data, conditioned on a textual prompt. To allow self- and cross-modality interaction, *joint attention* operates on concatenated text and image tokens. For editing, the condition image is concatenated to the noisy image tokens and processed through the same blocks as the noisy image (Tan et al., 2025; Labs et al., 2025; Wu et al., 2025). In this setting, the attention operator jointly processes *prompt*, *noisy latents*, and *context image* tokens. While this approach allows great global visual quality, allowing all tokens to interact with all other tokens causes inevitable semantic leakage as $N$ increases.

Formally, Flow matching learns a single global velocity field $v_\theta$ governing the evolution of all latents simultaneously. Conditioning is typically injected as a global signal, $v_\theta(x, t \mid c)$, without explicit mechanisms to enforce instance-level separation. As a result, when multiple instructions $\{c_n\}_{n=1}^N$ are applied simultaneously, their effects may interfere within the shared vector field, leading to semantic

---

[*]Equal contribution, order determined by coin flip  [1]UniMORE  [2]ETH Zurich  [3]Google. Correspondence to: C. Zaccagnino <czaccagnino@unimore.it>, E. Simsar <enis.simsar@inf.ethz.ch>.

*Proceedings of the 43rd International Conference on Machine Learning*, Seoul, South Korea. PMLR 306, 2026. Copyright 2026 by the author(s).

entanglement and unintended edits. This phenomenon, referred to as *attribute leakage* (Rassin et al., 2023; Chefer et al., 2023; Mun et al., 2025; Phung et al., 2024; Dahary et al., 2024), stems mainly from semantic interference in the prompt encoding (Zhou et al., 2024;?; Phung et al., 2024) and from the vanilla joint attention mechanism, which permits disjoint concepts to interact (Zhou et al., 2025).

To address this, we enforce instance independence *architecturally*. Recent works have demonstrated that reprogramming attention mechanisms at inference time can successfully enforce semantic guidance and structural constraints (Hertz et al., 2023; Chefer et al., 2023; Quattrini et al., 2024a;b). Building on this intuition, we introduce **Instance-Disentangled Attention**, a mechanism that partitions the joint attention operations within the MMDiT blocks (Esser et al., 2024) of the flow matching-based FLUX.1 Kontext model (Labs et al., 2025). By partitioning the attention, we enforce locality constraints without disrupting the global flow matching objective. Moreover, we can perform disentangled multi-edits in a single pass, greatly improving computational performance.

We propose to consider both natural image editing and text image editing on Infographic images. Current natural image multi-instance editing benchmarks (Chakrabarty et al., 2024) suffer from low-dimensionality and a low number of edits for each sample, with tens of images and just a few edits requested for each image. To address this, we chose to evaluate models on infographic text replacement. This domain has many practical applications, and is particularly challenging since infographics are text-dense and aesthetically-pleasing information visualizations (Mathew et al., 2022), where uncontrolled edits of a single instance can severely alter the semantics of the whole image. The strict spatial layout and semantic density of such visually rich documents have long posed distinct challenges for parsing and structural understanding (Lee et al., 2023; Blecher et al., 2024; Quattrini et al., 2024c). In the generative context, this domain facilitates semi-automatic high-quality annotation. Therefore, we curate a benchmark consisting of two sets of infographic images, paired with paragraph-level text region localization annotations and editing instructions in the form of *Change 'SRC' in 'TGT'*, where SRC is the original text (generally in English), and TGT is the desired text in another language. Our proposed benchmark contains infographic images of notable complexity, both in terms of the number of edits and the small area of the edited regions, opening up many challenges. The benchmark is available to the community[1]. Experimental analysis on these two domains demonstrates the effectiveness and efficiency of our approach.

---

[1] https://huggingface.co/datasets/
blowing-up-groundhogs/InfoEdBench

## 2. Related Work

**Generative Image Editing.** State-of-the-art editing has consolidated around Autoregressive models (Tian et al., 2024; Han et al., 2025; Team et al., 2025; Pippi et al., 2025; Zaccagnino et al., 2026) and Diffusion Transformers (Liu et al., 2025; Wu et al., 2025; Tan et al., 2025; Wang et al., 2025). While these architectures increasingly support multimodal conditioning, practical pipelines often struggle with fine-grained localization. To address this, recent works utilize attention regularization (Simsar et al., 2025; Zhang et al., 2025) or constrained cross-token interactions (Dahary et al., 2024; Huang et al., 2024) to improve spatial specificity. However, these methods primarily target single-region or global edits, often failing to maintain coherence when applied to multiple disjoint regions simultaneously.

**Multi-Instance Editing.** Simultaneously modifying multiple specific regions remains a significant challenge. Recent approaches mitigate interference via object-aware inversion (Yang et al., 2024), segmentation guidance (Matsuda et al., 2024), masked dual-editing (Zhu et al., 2025a), or conflict-aware multi-layer learning (Fu et al., 2026). Others specifically target quantity perception (Li et al., 2025) or localized multi-diffusion processes (Chakrabarty et al., 2024). However, these methods typically operate within the discrete denoising steps of diffusion models, relying on attention map alignments that break down under the high-density, structural constraints of infographics.

**Conditional Flow Matching.** Flow matching (FM) (Lipman et al., 2023) offers efficient inference and exact invertibility, serving as a powerful alternative to diffusion. Recent theoretical advances have further optimized these dynamics via optimal transport paths (Tong et al., 2024) and guidance mechanisms (Zheng et al., 2023). While other works investigate region-aware generation (Chen et al., 2025b; Eijkelboom et al., 2025) and causal masking (He et al., 2024) to inject spatial control, most FM editors (Labs et al., 2025; Esser et al., 2024) still rely on globally conditioned velocity fields. This global conditioning causes attribute leakage between regions during multi-instance editing (Zhou et al., 2025), a limitation we address via our instance-disentangled attention mechanism.

## 3. Proposed Approach

In this work, we devise an approach to address the problem of *multi-instance editing* under localized, potentially conflicting conditioning signals. The task translates into obtaining a conditional velocity field that transforms a source distribution (noise) into a target edited sample $x_1$, subject to a set of localized instructions $\{(s_n, b_n)\}_{n=1}^N$, where $s_n$ is a text prompt and $b_n$ is the localization information (*e.g.*, a bounding box) for the $n$-th instance. Without ex-

plicit structural constraints, the current generative models struggle to disentangle simultaneous, conflicting edits. This phenomenon, often referred to as *attribute leakage*, stems from the entangled nature of the cross-attention mechanism (Rassin et al., 2023; Chefer et al., 2023). Standard attention operations allow semantic tokens associated with one subject (*e.g.*, "red") to attend to spatial regions corresponding to another (*e.g.*, "car"), causing concepts to bleed across boundaries (Mun et al., 2025). While previous works have attempted to mitigate this via inference-time attention guidance (Hertz et al., 2023), these methods often require costly iterative optimization. Our approach[2] also works at inference time and addresses the root cause: the architectural permission for disjoint concepts to interact.

### 3.1. Preliminaries

**Conditional Flow Matching.** We focus on conditional rectified flow matching (Lipman et al., 2023; Tong et al., 2024) as the generative framework. Specifically, let $x_1 \sim p_{\text{data}}$ denote a target data sample and $x_0 \sim p_0$ a source sample drawn from a simple base distribution (*e.g.*, Gaussian noise). Flow matching learns a time-dependent velocity field conditioned on signals $\mathcal{C}$

$$v_\theta : \mathbb{R}^d \times [0,1] \times \mathcal{C} \to \mathbb{R}^d$$

such that the induced ODE

$$\frac{dx_t}{dt} = v_\theta(x_t, t \mid c)$$

transports $x_0$ to $x_1$, conditioned on the external information $c$. In the rectified flow formulation (Liu et al., 2023), the training objective minimizes the expected flow matching error over time $t \in [0,1]$, *i.e.*,

$$\mathcal{L}_{\text{FM}} = \mathbb{E}_{t,x_1,x_0} \left[ \|v_\theta(x_t, t \mid c) - (x_1 - x_0)\|^2 \right], \quad (1)$$

where $x_t = (1-t)x_0 + tx_1$. At inference time, samples are generated by integrating the learned ODE between 0 and 1.

**Joint Attention.** State-of-the-art conditional generative models based on flow matching (Esser et al., 2024; Labs et al., 2025) employ an MMDiT as the parameterization of $v_\theta$. MMDiT performs only self-attention operations on a sequence of tokens obtained by concatenating embeddings from multiple modalities. We denote the input tokens with $Z$, and with $Q$, $K$, $V$ the query, key, and value matrices linearly projected from $Z$ through learnable matrices. Specifically, let $Z = Z^{\text{text}} \parallel Z^{\text{latent}} \parallel Z^{\text{context}}$ denote the joint token sequence, where: (i) $Z^{\text{text}}$ encodes textual instructions, (ii) $Z^{\text{latent}}$ encodes the evolving conditional flow matching latent representation $x_t$, (iii) $Z^{\text{context}}$ encodes auxiliary conditioning signals (*e.g.*, a reference image, depth maps, or segmentation mask cues). Each attention layer within MMDiT applies *joint attention* (Esser et al., 2024),

---

[2]Code available at https://github.com/Blowing-Up-Groundhogs/IDAttn

where queries, keys, and values are computed independently for each modality and concatenated:

$$Q = Q^{\text{text}} \parallel Q^{\text{latent}} \parallel Q^{\text{context}},$$
$$K = K^{\text{text}} \parallel K^{\text{latent}} \parallel K^{\text{context}},$$
$$V = V^{\text{text}} \parallel V^{\text{latent}} \parallel V^{\text{context}}.$$

The attention output is then given by

$$\text{Attn}(Q, K, V) = \text{softmax}\left( \frac{QK^\top}{\sqrt{d}} \right) V. \quad (2)$$

### 3.2. Instance-Disentangled Attention

We introduce **Instance-Disentangled Attention** (IDAttn), an architectural intervention that enforces structured conditional dependencies within joint attention, enabling independent control of multiple instances within each single flow field estimation step.

**Token Space Partitioning.** First, consider $N$ instances to be edited within the same sample, *e.g.*, regions of a reference input image that the model should modify based on some textual prompt. Our approach entails logically partitioning the joint token sequence $Z$ into the following sets, according to both modality and instance association information (*e.g.*, bounding boxes $b_n$ in the case of image editing):

- $\mathbb{T}_g$: Global prompt tokens, *i.e.*, text tokens $Z^{\text{text}}$ relative to style, overall scene description, or a null prompt.
- $\mathbb{T}_n$: Local prompt tokens, *i.e.*, text tokens $Z^{\text{text}}$ specifically describing the target content for instance $n$.
- $\mathbb{L}_u$: Latent tokens $Z^{\text{latent}}$ not pertaining any instance (*e.g.*, relative to the background).
- $\mathbb{L}_n$: Local latent tokens, *i.e.*, latent tokens $Z^{\text{latent}}$ corresponding to the $n$-th instance.
- $\mathbb{C}_u$: Context tokens $Z^{\text{context}}$ relative to no specific instance. (*e.g.*, background in a reference image).
- $\mathbb{C}_n$: Local context tokens, *i.e.*, auxiliary conditioning tokens $Z^{\text{context}}$ relative to instance $n$.

Note that, when multiple instances overlap (*e.g.*, when localizing them via bounding boxes in the case of image editing), their respective tokens end up in multiple partitions.

**Disentanglement Attention Masks.** We govern the information flow via an additive attention mask $M \in \mathbb{R}^{|Z| \times |Z|}$, which we combine with the standard joint attention matrix. The attention operation for query $Q$, key $K$, and value $V$ in Equation (2) becomes:

$$\text{IDAttn}(Q, K, V, M) = \text{softmax}\left( \frac{QK^\top}{\sqrt{d}} + M \right) V, \quad (3)$$

where $M_{ij} \in \{0, -\infty\}$. We define two distinct masking regimes to control the entanglement of the latent space: one for instance disentanglement ($M^{\text{dis}}$), and the other for context-aware harmonization ($M^{\text{har}}$). Please refer to Figure 1 for a pictorial representation of the defined masks.

**Disentanglement Regime.** This regime enforces instance isolation. For any two distinct instances $n, m$ (where $n \neq m$), the mask is defined as:

$$M_{ij}^{\text{dis}} = \begin{cases} 0 & Z_i \in \mathbb{T}_g \cup \mathbb{L}_u \cup \mathbb{C}_u \wedge Z_j \notin \bigcup_{n=1}^{N} \mathbb{T}_n \\ 0 & Z_i, Z_j \in \mathbb{T}_n \cup \mathbb{L}_n \cup \mathbb{C}_n \quad \forall n \in [1, N] \\ -\infty & \text{otherwise.} \end{cases} \tag{4}$$

In other words, this mask lets the global prompt tokens $\mathbb{T}_g$ attend to all the image tokens ($\mathbb{L}_u \cup \{\mathbb{L}_n\}_{n=1}^{N}$) and auxiliary context tokens ($\mathbb{C}_u \cup \{\mathbb{C}_n\}_{n=1}^{N}$); the image tokens $\mathbb{L}_u$, and context tokens $\mathbb{C}_u$ relative to the background can attend to the global prompt $\mathbb{T}_g$ and to all the image tokens ($\mathbb{L}_u \cup \{\mathbb{L}_n\}_{n=1}^{N}$) and context tokens ($\mathbb{C}_u \cup \{\mathbb{C}_n\}_{n=1}^{N}$); the prompt tokens, image tokens, and context tokens relative to the same instance $n$ (*i.e.*, $\mathbb{T}_n$, $\mathbb{L}_n$, and $\mathbb{C}_n$) can only attend to each other. As a result, instance-specific conditioning tokens can influence only their corresponding image latents, while global tokens remain self-consistent.

For multi-instance image editing, the main effect of this masking strategy is enabling attention within the token subset corresponding to the region within $b_n$ and its textual editing prompt $s_n$, while preventing cross-instance interference between regions and editing prompts being ignored by the model. Moreover, by letting the global prompt and the background visual tokens attend to both each other and the instances' tokens, this masking allows maintaining visual consistency.

**Harmonization Regime.** To preserve global coherence, we relax the constraints defined in $M^{\text{dis}}$ by allowing the image and the auxiliary context tokens relative to each instance ($\mathbb{L}_n$, and $\mathbb{C}_n$) attend to also all the other image and context tokens ($\mathbb{L}_u \cup \{\mathbb{L}_n\}_{n=1}^{N}$, and $\mathbb{C}_u \cup \{\mathbb{C}_n\}_{n=1}^{N}$), further enforcing global coherence. In other words, the mask in this regime is defined as:

$$M_{ij}^{\text{har}} = \begin{cases} 0 & Z_i, Z_j \notin \bigcup_{n=1}^{N} \mathbb{T}_n \\ 0 & Z_j \in \mathbb{T}_n \cup \mathbb{L}_n \cup \mathbb{C}_n \wedge \\ & Z_i \in \mathbb{T}_n \quad \forall n \in [1, N] \\ -\infty & \text{otherwise.} \end{cases} \tag{5}$$

**Masks Application Strategy.** To obtain the edited sample, within each refinement step, we apply $M^{\text{har}}$ in the early and late layers of the velocity field estimation Transformer (denoted as $L_{\text{early}}$ and $L_{\text{late}}$, respectively), and $M^{\text{dis}}$ in the central layers ($L_{\text{mid}}$). The resulting instance-disentangled flow matching process is described in Appendix A. This design is motivated by empirical and theoretical analyses of Transformer representations (Xiang et al., 2023; Chen et al., 2025a; Yu et al., 2025), which suggest that early layers focus

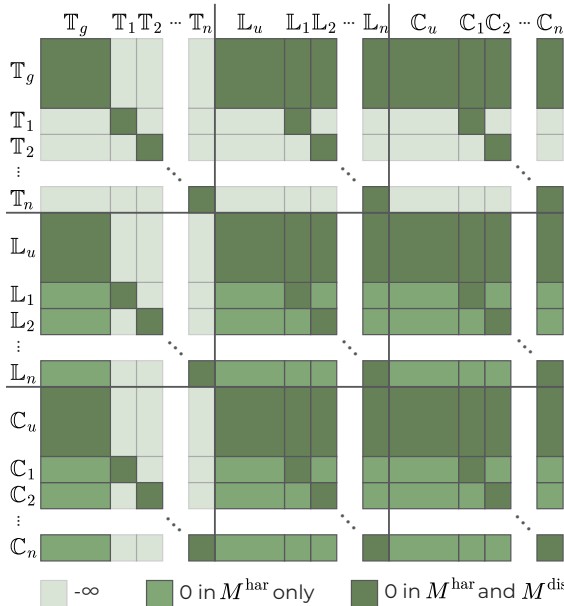

*Figure 1.* Logic visualization of the proposed joint attention masks.

on coarse feature extraction, middle layers encode semantic bindings, and late layers refine and globally harmonize the output. Similar layer-wise intervention strategies have proven effective in controlled generation (Zhou et al., 2025).

### 3.3. Efficient Multi-Prompt Independent Encoding

Effective multi-instance editing requires ensuring that textual instructions for different instances remain semantically isolated. Standard approaches typically concatenate all instructions into a single long prompt. However, this allows the text encoder's self-attention mechanism to mix unrelated concepts before they even reach the generative model (Chefer et al., 2023; Rassin et al., 2023).Existing solutions generally fall into the following two categories. (i) *Post-encoding single prompt masking* (Xie et al., 2023), *i.e.*, encoding a single prompt and enforcing spatial constraints by masking parts of the encoded prompt within the attention layers of the generative architecture. This approach is efficient in terms of token usage, but instance-level disentanglement is not guaranteed and often worsens as the prompt complexity increases. (ii) *Multi-prompt independent encoding* (Zhou et al., 2025), *i.e.*, dividing the prompt into instance-specific subprompts before encoding, each of which is padded or truncated to a fixed length. This enforces instance-level separation by construction, but its computational cost scales linearly with the number of instances.

We propose a multi-prompt encoding strategy that ensures semantic isolation by construction while maintaining a sequence length proportional to the semantic content, thereby optimizing the computational cost of the subsequent attention operations. First, we decompose the input prompt into a global component $s_g$ (in practice, we use a null prompt) and

independent instance-specific components $\{s_n\}_{n=1}^N$. Then, we encode these components separately to obtain variable-length embeddings, and concatenate them to construct the final conditioning sequence.

### 3.4. Domain-Specific Fine-Tuning

All of the modifications proposed so far can be applied at inference time on top of any pre-trained model without additional training. However, state-of-the-art flow matching models for image generation and editing are typically trained on natural images with long prompts acting on relatively large areas. As a result, they may underutilize fine-grained conditioning signals corresponding to small spatial regions or short textual instructions. Even though our method is primarily designed to operate at inference time and improves upon the baseline even in these out-of-domain settings, we further explore whether a lightweight domain adaptation step with our proposed modifications would improve its performance in these cases. We perform supervised adaptation on the paired subset of the *Crello Edit* benchmark introduced in Section 4. Specifically, we apply Low-Rank Adaptation (Shen et al., 2022) with rank of $r = 32$ to all layers of the MMDiT, once equipped with the proposed joint attention masking strategy. We minimize the conditional flow matching loss $\mathcal{L}_{\text{FM}}$ defined in Equation (1), where the text input is handled with our multi-prompt independent encoding strategy. This adaptation step is optional and incurs a modest computational overhead. In particular, domain-specific fine-tuning significantly enhances the model's ability to preserve typography, style, and visual coherence, serving as a robust extension for highly complex, multi-instance domains. We further explore this aspect in Appendix G.

## 4. Infographics Editing Benchmark

While recent benchmarks have explored object-level and region-based image editing, none explicitly address scenarios requiring a large number of fine-grained edits to be applied simultaneously within a single image. Current multi-instance benchmarks remain limited in both the number of editable regions per image and overall scale (Chakrabarty et al., 2024; Yang et al., 2024), arguably due to the cost of accurate annotation in natural images. As a result, they fail to stress-test the ability of editing models to maintain edit disentanglement and locality as the number of concurrent instructions increases. In this respect, we argue that infographics constitute a particularly suitable domain for studying this setting, due to the high density of visual and textual elements within a coherent layout that characterizes these images. In this work, we focus on textual elements, which constitute a demanding and practically relevant testbed for evaluating multi-instance image editing methods. In fact,

since infographics often contain a large number of text boxes ($N \gg 5$), when attempting end-to-end editing of all boxes in one go, or in multi-turn, existing image editing models tend to produce inconsistent or collapsed edits and to ignore most of the editing prompts. Moreover, text boxes are usually small relative to the full image, implying that the model must modify high-frequency, localized details. Nonetheless, this setting allows for scalable data collection. In fact, text regions can be automatically detected and annotated with standard Text Detection and Optical Character Recognition (OCR) pipelines, making it feasible to construct large datasets efficiently. Therefore, in this work, we devise a large collection of densely-annotated infographic text editing samples. This will be made publicly available to favor reproducibility and exploration of the task at hand.

Formally, let $I^{\text{ref}} \in \mathbb{R}^{H \times W \times 3}$ denote the high-resolution input infographic image and consider a set of $N$ bounding boxes $B = \{b_n = (x_n, y_n, w_n, h_n)\}_{n=1}^N$, each enclosing a text element, and a set of corresponding editing instructions $S = \{s_n\}_{n=1}^N$, *e.g.*, *Change 'Hello' to 'Bonjour'*. In real scenarios, these latter two inputs can either be provided by a user or by auxiliary modules for text detection, recognition, and translation. The task entails generating an edited image $I^{\text{edit}}$ such that each text element within $b_n$ is replaced according to $s_n$ while maintaining visual coherence of $I^{\text{ref}}$ and text style within each local edit. We devise the following subsets of increasing level of difficulty.

**Crello Edit.** To obtain the most manageable subset, we start from the layer-wise annotated graphic designs of Crello (Yamaguchi, 2021). These are mostly images with English text. Crello comes with precise annotations of each text box, which we translate into four languages (French, German, Italian, and Spanish) by using Seed-X (Cheng et al., 2025). Then, we re-render the text layer of the original sample so that it contains the translated text, by using the same font metadata accompanying it. In this way, we obtain a paired editing dataset featuring samples where the reference image contains English text and the target image contains text in another language, which can be used for supervised training. To reduce artifacts, we only re-render text boxes where the length of the translated text is up to 25% longer than the original one, and inspect the samples for visual quality.The obtained set contains 1512 samples that can be used in the training phase and 4367 samples used for test, accompanied by metadata to obtain 1 to 25 editing instructions on boxes whose area is on average 4.23±5.49% of the entire image.

**InfoEdit.** To explore a more challenging and realistic infographic editing scenario, we start from the images collected by the authors of (Zhu et al., 2025b) and retain only the real, professionally-made, and open-source English infographics. We consider images whose total area is at least of $1024^2$px and further refine the selection via visual inspection. Then,

*Table 1.* Ablation study on layer scheduling on LoMOE-Bench.

| $L_{early}$ | $L_{mid}$ | $L_{late}$ | Tgt CLIP↑ | Bg LPIPS↓ | Loc CLIP↑ | AR [%]↑ |
|---|---|---|---|---|---|---|
| $M^{dis}$ | $M^{dis}$ | $M^{dis}$ | 25.51 | 0.108 | 29.14 | 94.27 |
| $M^{dis}$ | $M^{har}$ | $M^{har}$ | 25.01 | 0.099 | 28.53 | 82.29 |
| $M^{dis}$ | $M^{dis}$ | $M^{har}$ | 25.63 | 0.100 | 29.21 | 91.15 |
| $M^{dis}$ | $M^{har}$ | $M^{dis}$ | 25.03 | 0.097 | 28.62 | 80.21 |
| $M^{har}$ | $M^{har}$ | $M^{dis}$ | 25.11 | 0.101 | 28.63 | 83.33 |
| $M^{har}$ | $M^{dis}$ | $M^{dis}$ | 25.59 | 0.103 | 29.23 | 93.23 |
| $M^{har}$ | $M^{dis}$ | $M^{har}$ | **25.67** | **0.091** | **29.26** | 92.19 |
| $M^{har}$ | $M^{har}$ | $M^{har}$ | 25.05 | 0.103 | 28.54 | 80.21 |

*Table 3.* Quantitative results on LoMOE-Bench.

| | Tgt C↑ | SD↓ | IR↑ | HPS↑ | LPIPS$_B$↓ | SSIM$_B$↑ | Loc C↑ | AR$_%$↑ |
|---|---|---|---|---|---|---|---|---|
| **LoMOE** | **26.00** | **0.067** | 0.266 | 0.546 | **0.090** | 0.834 | **29.40** | 98.96 |
| **LayerEdit** | 25.61 | 0.071 | 0.252 | 0.186 | 0.147 | 0.864 | 29.07 | 100.00 |
| **FLUX** | 24.71 | 0.078 | 0.266 | -0.059 | 0.206 | 0.830 | 27.58 | 94.79 |
| **FLUX $\mu$T** | 25.71 | 0.074 | **0.282** | 0.550 | 0.150 | 0.873 | 28.27 | 94.27 |
| **FLUX w/ v.c.** | 24.49 | 0.072 | 0.267 | 0.265 | 0.170 | 0.893 | 27.60 | 92.71 |
| **Ours** | 25.60 | 0.073 | 0.277 | **0.574** | 0.099 | **0.919** | 29.08 | 89.06 |

*Table 2.* Ablation study on prompt encoding and application of instance-disentangled attention on LoMOE-Bench.

| Efficient prompt enc. | IDAttn | Tgt C↑ | LPIPS$_B$ ↓ | Loc C↑ | AR$_%$ ↑ |
|---|---|---|---|---|---|
| | | 25.22 | 0.126 | 27.93 | 86.46 |
| ✓ | | 25.16 | 0.129 | 27.90 | 86.46 |
| | ✓ | **25.67** | **0.091** | **29.26** | 92.19 |
| ✓ | ✓ | 25.60 | 0.099 | 29.08 | 89.06 |

we run LayoutParser (Shen et al., 2021) to detect the text elements and PaddleOCR (Cui et al., 2025) to transcribe them. Finally, we aggregate the lines into paragraphs when relevant. Also in this case, we resort to Seed-X (Cheng et al., 2025) to obtain the translations of the text elements into French, German, Italian, and Spanish, and obtain the located editing prompts from those. The resulting dataset contains 4960 samples, where images are paired with 1 to 285 editing instructions relative to boxes whose area is on average 0.62±1.64% of the entire image.

## 5. Experimental Analysis

**Considered Datasets.** We evaluate our approach on multi-instance editing of both natural images and infographics. For the former case, we resort to the recently-proposed LoMOE-Bench (Chakrabarty et al., 2024), featuring 80 images paired with 2 to 7 editing instructions. For the latter, we use the Crello Edit and InfoEdit test sets featured in our proposed Infographic Editing Benchmark (Section 4).

**Competitors and Baselines.** To assess the benefits of our proposed approach for multi-instance image editing, we apply it to the State-of-the-Art FLUX.1 Kontext FM-based generative model (Labs et al., 2025) and compare its performance against the following variants, officially proposed by the respective authors: (i) we apply FLUX.1 Kontext end-to-end by prompting it with all the text editing prompts in one go (**FLUX**); (ii) we apply it in a multi-turn scenario, where we provide it with one editing prompt at a time (**FLUX $\mu$T**). Moreover, for natural image editing, we also consider **Lo-MOE** (Chakrabarty et al., 2024) and **LayerEdit** (Fu et al., 2026) and an additional variant of FLUX.1 Kontext, described in (Labs et al., 2025) that entails drawing boxes over the editing instances as visual cues, and referring to them

by color in the prompt (**FLUX w/ v.c.**). As for infographics text editing, we include a FLUX.1 Kontext-based baseline that we design for ensuring instance disentanglement and isolation. We use the model for editing each text box separately, and we stitch the box back on the original image, after having run the ViTEraser (Peng et al., 2024) inpainting model for removing the original text (**FLUX st.**). More-over, we consider **Calligrapher** (Ma et al., 2025), which is a finetuned version of FLUX-Fill (Labs et al., 2025) for text image editing. Since it can handle one editing instruction at a time, we use it in a multi-turn scenario (**Calligrapher $\mu$T**) and in the same box editing and re-stitching pipeline as for FLUX.1 Kontext (referred to as **Calligrapher st.**).

**Considered Scores.** Here we mention the scores we use to evaluate the performance of our approach and the competitors (further details are given in Appendix D). For our Infographics Editing Benchmark, we propose to use the following scores: the average Character Error Rate (**CER**) obtained by PaddleOCR over all edited text boxes, and the difference with the CER obtained on the ground-truth (**ΔCER**) for the Crello Edit subset; the **MAE** and **MSE** on the background pixels, which should remain untouched after editing (**MAE$_B$** and **MSE$_B$**); the Fréchet Inception Distance (**FID**) (Heusel et al., 2017); and the Attempt Rate (**AR$_%$**), which we introduce to estimate the percentage of boxed that the model has attempted to edit. For LoMOE-Bench (Chakrabarty et al., 2024), we use the scores proposed by its authors, *i.e.*,: Target CLIP Score (**Tgt C**) (Hessel et al., 2021), Structural Distance (**SD**), Image Reward (**IR**) (Xu et al., 2023) and Human Preference Score (**HPS**) (Wu et al., 2023) on the entire image; LPIPS (**LPIPS$_B$**) (Zhang et al., 2018) and SSIM (**SSIM$_B$**) (Wang et al., 2004) on the background pixels. We also use **FID** and **AR$_%$**, and the CLIP Score on the edited regions (**Loc C**).

Furthermore, we conduct a **user study** on 40 users, asking them to answer 108 binary choice questions on the three datasets, comparing our approach, FLUX-K and FLUX-K $\mu$T. Specifically, we request to pick the best output between two displayed images from two random models and samples, based on the given source image and edit instruction. Additionally, we prompt **Gemini** 3 Flash in an LLM-as-a-judge setting on the same set of questions. We report results for both these evaluations in terms of Elo score.

*Table 4.* Quantitative results on the Infographic Editing Benchmark.

| | **Crello Edit** | | | | | | **InfoEdit** | | | | |
| --- | --- | --- | --- | --- | --- | --- | --- | --- | --- | --- | --- |
| | FID↓ | MAE$_B$↓ | MSE$_B$↓ | CER↓ | ΔCER↓ | AR [%]↑ | FID↓ | MAE$_B$↓ | MSE$_B$↓ | CER↓ | AR [%]↑ |
| Calligrapher $\mu$T | 10.15 | 3.82 | 8.58 | 0.73 | 0.29 | 51.21 | 113.23 | 40.34 | 69.80 | 0.92 | 99.98 |
| Calligrapher st. | 14.40 | **2.06** | **8.46** | 0.69 | 0.24 | 99.47 | 13.37 | 39.09 | 70.72 | 0,82 | 99.31 |
| FLUX | 10.06 | 61.83 | 91.73 | 0.65 | 0.20 | 68.72 | 4.36 | 30.20 | 64.26 | 0.77 | 39.94 |
| FLUX $\mu$T | 12.10 | 62.13 | 91.32 | 0.63 | 0.19 | 90.44 | 65.73 | 35.17 | 66.76 | 0.90 | 99.81 |
| FLUX st. | 15.69 | 61.33 | 90.90 | 0.59 | 0.15 | 73.49 | 10.48 | 29.46 | 62.67 | 0.66 | 63.25 |
| Ours | **9.45** | 61.72 | 91.58 | 0.61 | 0.16 | 52.00 | **2.41** | 3.41 | 15.33 | 0.64 | 52.61 |
| Ours + ft | 10.85 | 61.35 | 90.86 | **0.52** | **0.07** | 92.16 | 2.80 | **3.22** | **13.41** | **0.56** | 80.90 |

## 5.1. Results

In Table 1, we report an ablation study, performed on LoMOE-Bench, on the effect of applying the proposed masks at different stages of the MMDiT. CLIP scores and LPIPS$_B$ indicate that our proposal, balancing harmonization in the early and late layers with disentanglement in the middle layers, better adheres to the prompt and preserves the background. Moreover, Table 2 contains an ablation analysis on the effect of our proposed instance-disentangled attention and the efficient prompt encoding strategy. While the former improves all the scores, efficient prompt encoding slightly lowers them. However, the main benefit of this strategy is in terms of efficiency, and is the only feasible option when editing images with tens or hundreds of instances (see Figure 3). Additional scores are in Appendix E.

A quantitative comparison of the considered models on LoMOE-Bench is reported in Table 3. We observe that the end-to-end FLUX baseline has the poorest performance, improved by the multi-turn variant. This indicates that, even with 2-7 edits, FLUX is unable to follow all of the prompts, but is comparatively better at following all of the individual prompts. The addition of visual cues in the form of bounding boxes superimposed on the image does not help in this regard. On the other hand, our approach exhibits solid performance, often ranking best or second-best.

The quantitative results on the Infographics datasets are reported in Table 4. We can see how the single-pass FLUX baseline shows high text rendering error and cannot follow the majority of the editing instructions, with just 40% AR$_\%$ on InfoEdit. On the other hand, our approach, and its fine-tuned version (**ft**), performs better on most scores, with a notable improvement in terms of CER and AR$_\%$ indicating how the model is attempting many more edits which are, on average, more correct. Moreover, the lower background scores indicate the absence of artifacts in those regions. Interestingly, these scores are comparable with those of FLUX st, which does not have background artifacts by design. Note that, by design, the stitching models also have a higher AR$_\%$, as each edit is performed separately and stitched upon the inpainted background, resulting in a false-positive.

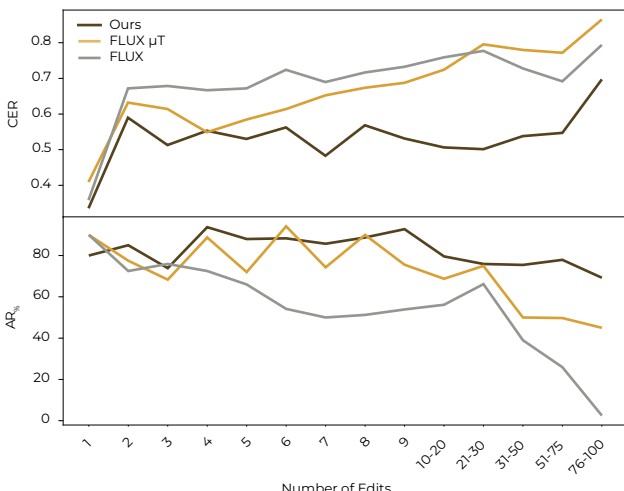

*Figure 2.* CER and AR *w.r.t.* the number of edits.

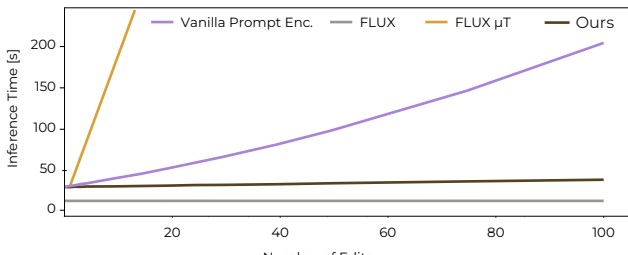

*Figure 3.* Inference time comparison *w.r.t.* the number of edits.

Additionally, we analyze how editing capabilities scale when increasing the number of edits in Figure 2. We group samples in our Infographics benchmark by number of edits, with up to 20 randomly selected samples per group, and compute the average CER and AR$_\%$. We can see that our model scales better even on samples with tens of editing instances, while other methods ignore the editing instructions and commit more text rendering mistakes.

The results of our user study and the LLM-as-Judge evaluation are reported in Table 5. We observe that both the users and Gemini strongly favor our method, with the FLUX end-to-end baseline being the second-best. Noting the strong correlation between the automated LLM-as-a-judge results and the user study, we report results of an extended LLM-

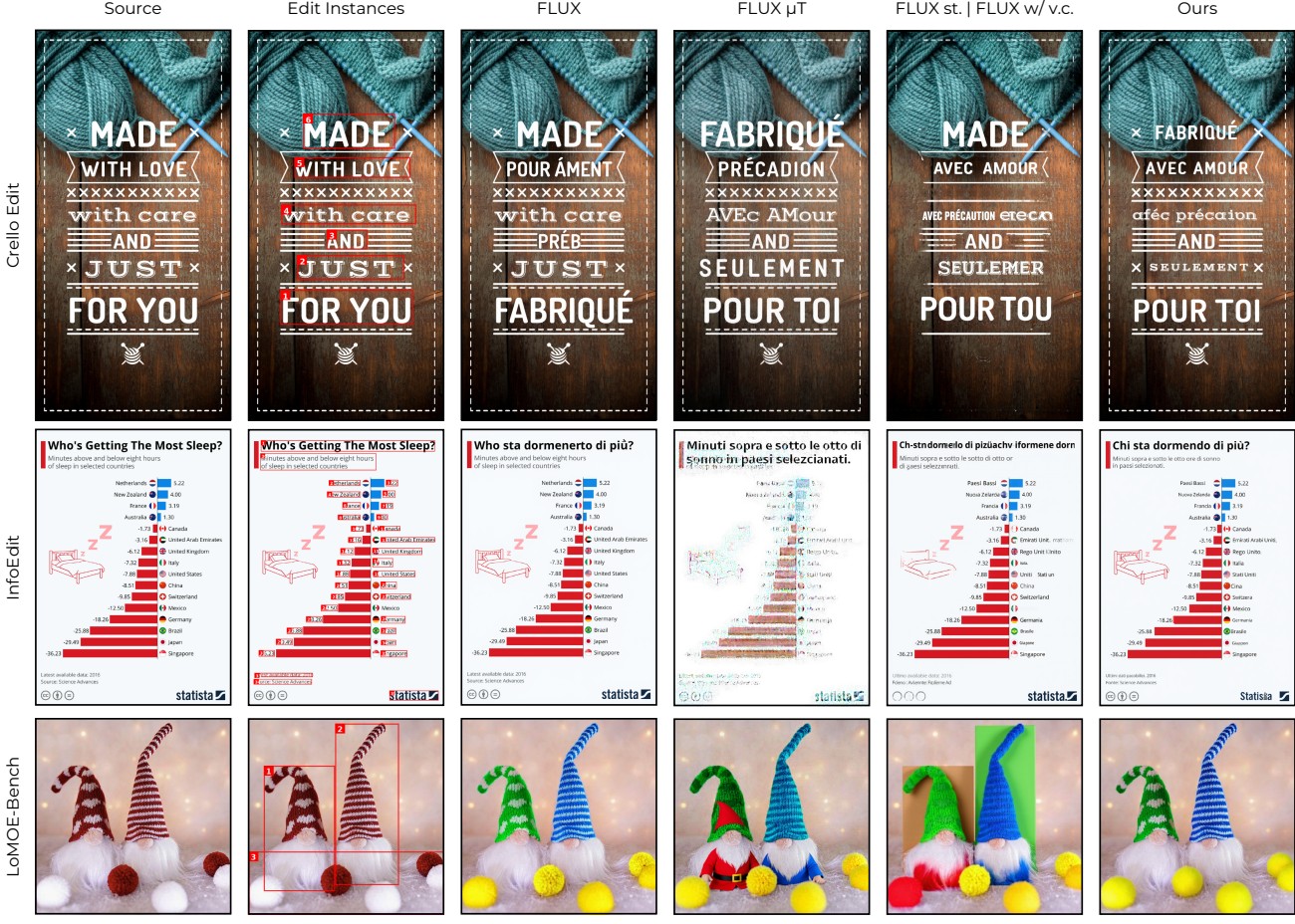

*Figure 4.* Qualitative results on **Crello Edit**, **InfoEdit** and **LoMOE-Bench**. The leftmost column is the source image, the second one is the source image with overlayed bounding boxes indicating areas in which to perform editing operations, and the other columns contain qualitative results on that same sample. For readability, we list the prompts for these samples in Appendix H.

*Table 5.* Elo scores of our approach on natural images and Infographics with a User Study and LLM-as-a-judge.

| | LoMOE-Bench | | Infographics | |
|---|---|---|---|---|
| | Users | Gemini | Users | Gemini |
| **FLUX** | 1331 | 1319 | 1102 | 1109 |
| **FLUX $\mu$T** | 680 | 695 | 1095 | 1096 |
| **Ours** | **1589** | **1585** | **1404** | **1394** |

as-a-judge evaluation with more evaluated baselines and samples in Appendix F.

Finally, in Figure 4, we present qualitative results on samples from the three datasets. In the first image, which has 6 editing instructions, the end-to-end FLUX baseline ignores half the prompts and incorrectly edits an area (where the word *AND* appears) despite the editing instruction entailing rewriting *AND* there. The multi-turn and stitching variants attempt all edits but make several mistakes, while our approach offers a good balance between editing quantity and quality. On the sample from InfoEdit, the FLUX only attempts one edit in the large title area with some rendering

mistakes. Due to the high number of iterations, FLUX $\mu$T severely degrades image quality. FLUX st. fares somewhat better by attempting all edits with higher success, but introduces many artifacts and character errors. Our method is capable of editing almost all of the instances, introducing minor artifacts, and only missing a few characters for one of the text rewriting prompts. On LoMOE-Bench, FLUX performs the requested edits but renders slightly unrealistic hats with no shadows. The multi-turn variant introduces artifacts and extra textures, which are even more evident for the variant exploiting visual cues (it draws colored rectangles, disrupting visual coherence and fidelity to the source image). Our approach successfully performs all of the requested edits. The prompts used to generate these samples are listed in Appendix H, for readability, and more qualitative results for all three datasets are shown in Appendix I.

## 5.2. Edge Cases and Limitations

We observe that our proposed attention biasing mechanism is robust to imperfect localization. We argue that this is due

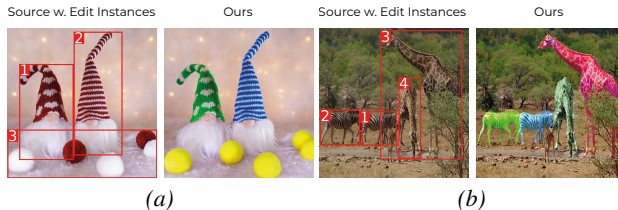

*(a)*       *(b)*

*Figure 5.* Qualitative results of our proposed IDAttn applied with imprecise (a) and overlapping (b) bounding boxes. For readability, we list the prompts for these samples in Appendix H.

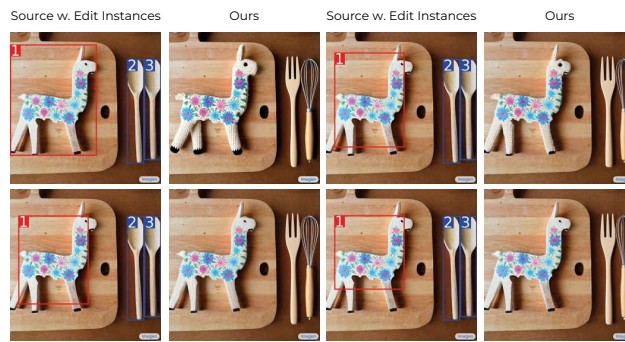

*Figure 6.* Qualitative results of our proposed IDAttn applied with imprecise and tight bounding boxes. For readability, we list the prompts for these samples in Appendix H.

to the fact that, even though we restrict the context in most layers to the instance-specific bounding boxes, the backbone model can still rely on its original localization capabilities within the instance's box (as an example, please refer to the broad bounding box relative to the cotton balls in Figure 5a). As for cases where bounding boxes fully overlap (*e.g.*, two giraffes in Figure 5b), these capabilities are enhanced by our biasing strategy coupled with the normalization intrinsic to attention. In such cases, IDAttn allows for more intense interactions within smaller instances, due to the fact that the softmax of the attention scores computed over a smaller region is less diluted by attention performed on background pixels when compared to what happens when computing the softmax on larger regions. This creates an implicit focus on the editing instruction that refers to the smaller box for the area in common between two overlapping instructions, which is usually a beneficial side product. Therefore, in the case of nested boxes, the edit for the smaller region is performed in case of conflicts, as in Figure 5b: prompt 3 (asking to change the color to yellow and pink) insists on the same area as prompt 4 (asking to change the color to white and green), and both edits are correctly applied.

A limitation of our approach for multi-instance editing is that, like previous approaches (Chakrabarty et al., 2024; Fu et al., 2026), it relies on external inputs to localize instances. Nonetheless, devising an end-to-end approach that can localize and apply the prompt-specified edits is a challenging goal beyond the scope of this work. Here, we focus on the editing component, which remains challenging even when perfect localization is provided. This is certainly an interesting future direction, and we argue that it could reasonably be tackled, *e.g.*, with agentic pipelines orchestrating state-of-the-art promptable detection or segmentation approaches for natural images, and text detection and OCR approaches for infographics, in addition to MMDiT-based editing models equipped with strategies for multi-instance concurrent editing such as ours. Note that editing failures can occur when highly inaccurate or completely wrong boxes are provided. However, our approach is robust to imprecise localization (*i.e.*, too tight or too broad boxes, as those in Figure 6 and in Figure 5) and can still reasonably identify the instance of interest. As for possible artifacts caused by hard truncation when localizing using bounding boxes, these are prevented by the harmonization biasing regime (*e.g.*, in the

bottom-right of Figure 6, the crochet and wooden textures are blended rather than abruptly interrupted).

## 6. Conclusion

In this work, we have explored the challenges of multi-instance editing within the framework of Flow Matching and MMDiTs. We have identified that the reliance on global velocity fields and joint attention mechanisms can lead to semantic entanglement when multiple instructions are processed concurrently. To address this, we have proposed Instance-Disentangled Attention, a modification that seeks to localize attention operations by associating specific textual tokens with their corresponding spatial regions. Moreover, we have devised a multi-prompt encoding strategy that makes inference more efficient. Our experimental results on natural images and our devised benchmark of challenging infographic text editing suggest that our approach can help mitigate attribute leakage and improve prompt following ability while maintaining global coherence. Furthermore, by performing edits in a single inference pass, the method offers a more efficient alternative to multi-step or iterative editing pipelines as the number of instances increases. Finally, it is worth noting that, while our implementation was integrated into the FLUX architecture, the concept of attention partitioning applies straightforwardly to other Transformer-based flow models, particularly those based on MMDiT.

## Impact Statement

This paper presents work aimed at advancing multi-instance text-guided image editing. Our approach facilitates the rapid localization and updating of visual information, potentially lowering barriers to cross-cultural communication by enabling high-fidelity text translation within strict layouts. Furthermore, by optimizing for single-pass editing, our method offers improvements in computational efficiency compared to iterative frameworks. There are many other potential societal consequences of our work, none which we feel must be specifically highlighted here.

## Acknowledgment

This paper is based upon work supported by the GCP Credit Award, the Google Cloud Research Credits program with the award GCP19980904, and the FARD2025 project (CUP E93C25000370005). We acknowledge EuroHPC Joint Undertaking and ISCRA for awarding us access to LUMI at CSC, Finland, LEONARDO at CINECA, Italy, and MareNostrum5 at BSC, Spain.

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

# A. Masking Application Strategy

In Algorithm 1 we share the full algorithm of our multi-instance editing method. In particular, it shows how and where standard rectified flow inference is adapted to implement Instance-Disentangled Attention.

---

**Algorithm 1** Flow Matching-based Editing with Instance-Disentangled Attention

---

**input** Initial latent $x_0 \sim p_0$; **Conditioning** $Z^{\text{text}}, Z^{\text{context}}$; **Partitions** $\{\mathbb{T}_g, \mathbb{T}_n, \mathbb{L}_u, \mathbb{L}_n, \mathbb{C}_u, \mathbb{C}_n\}_{n=1}^N$; Steps $T$
**output** Edited sample $x_1$
1: {Construct masks using definitions in Equation (4) and Equation (5)}
2: $M^{\text{dis}}, M^{\text{har}} \leftarrow \text{ConstructMasks}(\{\mathbb{T}, \mathbb{L}, \mathbb{C}\})$
3: Initialize $x \leftarrow x_0, \Delta t \leftarrow 1/T$
4: **for** $k = 0$ to $T - 1$ **do**
5:     $t \leftarrow k \cdot \Delta t$
6:     $Z^{\text{latent}} \leftarrow \text{ImageEncoder}(x)$
7:     $Z \leftarrow Z^{\text{text}} \parallel Z^{\text{latent}} \parallel Z^{\text{context}}$
8:     $h \leftarrow \text{Embed}(Z, t)$
9:     **for** $\ell = 1$ to $L$ **do**
10:        **if** $\ell \in L_{\text{early}} \cup L_{\text{late}}$ **then**
11:           $M \leftarrow M^{\text{har}}$
12:        **else**
13:           $M \leftarrow M^{\text{dis}}$
14:        **end if**
15:        $h \leftarrow \text{TransformerBlock}(h, \text{mask} = M)$
16:     **end for**
17:     $v_\theta \leftarrow \text{OutputHead}(h)$
18:     $x \leftarrow x + v_\theta \cdot \Delta t$
19: **end for**
20: **return** $x$

---

# B. Implementation Details

We implement and analyze our proposed approach on the FLUX.1 Kontext (Labs et al., 2025) image editing framework. It uses a custom VAE as an image encoder, and a custom MMDiT (Esser et al., 2024) as a generative model. In all our experiments we adopt the same layer scheduling setting proposed in (Zhou et al., 2025), using layer 10 as the cut-off from $L_{\text{early}}$ to $L_{\text{mid}}$ and layer 47 as the cut-off from $L_{\text{mid}}$ to $L_{\text{late}}$. We argue that these cut-offs are appropriate to ensure that the majority of layers run in the disentanglement regime, and only the first and last 10 layers run in the harmonization regime to ensure the build-up of a coherent latent representation of the image within the MMDiT, and to harmonize the content to avoid artifacts due to the mask cut-off.

The instance prompt tokens $\{\mathbb{P}_n\}_{n=1}^N$ are obtained by encoding the instance text instructions $\{s_n\}_{n=1}^N$ using its finetuned T5 (Raffel et al., 2020) text encoder, where the effective tokens are those obtained until the EOS token is generated. The entire $Z^{\text{context}}$ sequence is obtained by encoding the input image with FLUX's custom VAE, whereas the $Z^{\text{latent}}$ is initialized to Gaussian noise. The instance tokens $\{\mathbb{L}_n\}_{n=1}^N \{\mathbb{C}_n\}_{n=1}^N$ are obtained from $Z^{\text{latent}}$ and $Z^{\text{context}}$, respectively, by converting the $\{b_n\}_{n=1}^N$ bounding boxes to masks. Specifically, the instance latent and context tokens retain the place they have within the context image's token sequence $Z^{\text{context}}$ as encoded by the VAEs, and the mask acts on those areas of the sequence. As global prompt $\mathbb{P}_g$ we use a null prompt made up of 200 padding tokens, which we call a *utility prompt*: it is the only prompt that interacts with $\mathbb{L}_u$ and $\mathbb{C}_u$ in the layers using $M^{\text{dis}}$, so it is needed to ensure effective background preservation.

# C. Attention Masking Implementation Analysis

Here we report a detailed analysis of the attention masking approach when applied to FLUX.1 Kontext as described in Appendix B. In particular, considering the way the inputs are encoded by the Flux VAE, and our efficient text encoding scheme on top of T5, we discuss the attention mask values as they appear when running the model on an image from our benchmark datasets.

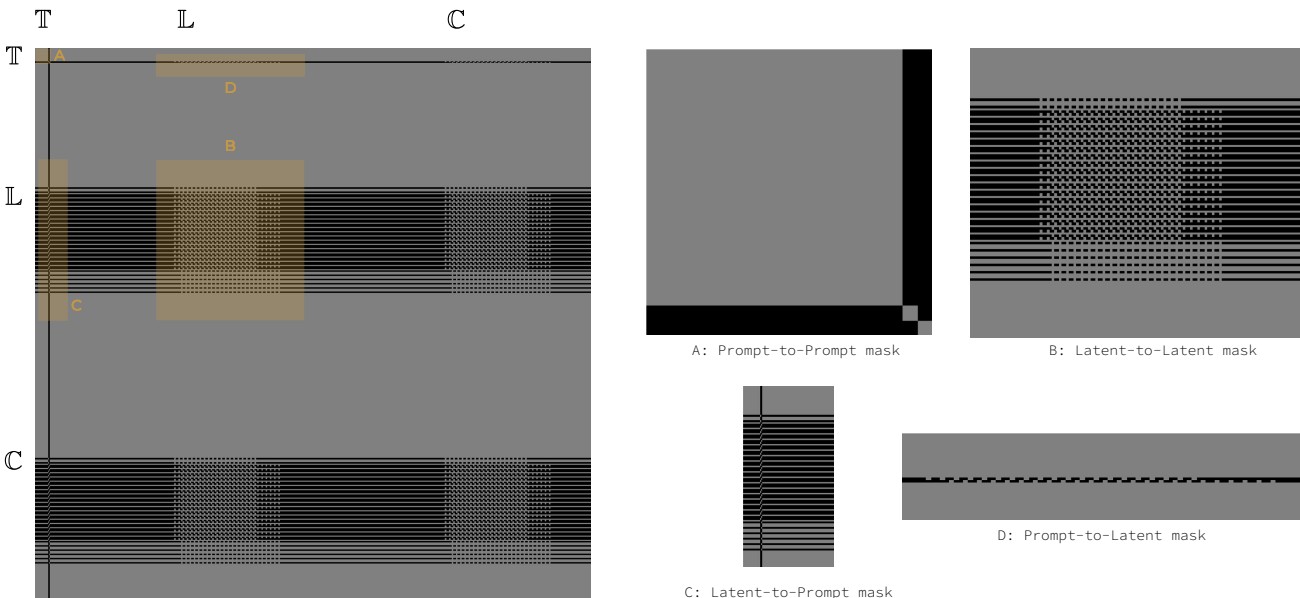

*Figure 7.* Detailed representation and analysis of the construction of $M^{\text{dis}}$ when using FLUX Kontext as a baseline on a sample from LoMOE-Bench. We use the color black to indicate areas that are blocked from attending to each other and the color grey to indicate areas where attention is allowed.

In this section we use the following definitions for simplicity of notation and as a direct reference to our specific implementation on FLUX.1 Kontext for multi-instance image editing:

$$\mathbb{T} := Z^{\text{text}} = \mathbb{T}_{\text{g}} \parallel \mathbb{T}_1 \parallel ... \parallel \mathbb{T}_{\text{N}}$$

$$\mathbb{L} := Z^{\text{latent}} = \mathbb{L}_{\text{u}} \parallel \mathbb{L}_1 \parallel ... \parallel \mathbb{L}_{\text{N}}$$

$$\mathbb{C} := Z^{\text{context}} = \mathbb{C}_{\text{u}} \parallel \mathbb{C}_1 \parallel ... \parallel \mathbb{C}_{\text{N}}$$

In Figure 7 and Figure 8 we respectively show the $M^{\text{dis}}$ and $M^{\text{har}}$ attention masks constructed for a sample from LoMOE-Bench, where each pixel is a query-key combination in the attention matrix, each row in the image is a different query, each column is a key, so by looking at a row we can see to which tokens of the sequence each query attends to.

In Figure 7 we can see, from top to bottom, the relatively large $\mathbb{T}_{\text{g}}$ (200px high) being allowed to attend to everything, except for the thin strip where the keys of the two instance prompts $\mathbb{T}_1$ and $\mathbb{T}_2$ are placed within the sequence. Below that, the horizontal strip corresponding to those two instance prompts should be analyzed in detail: the first 12 rows of pixels are those of the queries derived from $\mathbb{T}_1$, which do not attend to the global prompt, attend to the 12 columns of pixels referring to $\mathbb{T}_1$'s keys. This interactions among tokens in $\mathbb{T}$ is shown within Figure 7 in the zoomed-in section on the right called *A: Prompt-to-Prompt mask*.

Proceeding along the $\mathbb{T}_1$ row, the queries are allowed to attend also to the regions within the latent $\mathbb{L}$ and the context $\mathbb{C}$ that fall within the bounding boxes associated with the first instance prompt: this is the upper row of rectangles within *D: Prompt-to-Latent mask*. They look like isolated regions, because the tokens are encoded in raster order, and the box is not as wide as the entire image, but much smaller. The next 11 rows of pixels are those controlling attention masking for the second prompt, for which the same considerations apply, but in reference to a different instance.

The areas corresponding to the $\mathbb{L}$ and $\mathbb{C}$ tokens are exactly the same as each other because the positions of the bounding boxes within the latents and within the context are the same for this task. In *C: Latent-to-Prompt mask* the interaction between the queries of the latent and the keys of the prompt is highlighted: the tokens attending to the global prompt and all of the latents and context tokens are those corresponding to the background $\mathbb{L}_u$ and $\mathbb{C}_u$, whereas those corresponding to the latent instances $\mathbb{L}_n$ and context instances $\mathbb{C}_n$ only attend to their corresponding $\mathbb{P}_n$, $\mathbb{L}_n$ and $\mathbb{C}_n$, as shown in particular in *B: Latent-to-Latent mask*.

$M^{\text{har}}$, shown in Figure 8, is a relaxation of $M^{\text{dis}}$ allowing all instance tokens $\mathbb{L}$ and context tokens $\mathbb{C}$ to attend to all instance and context tokens.

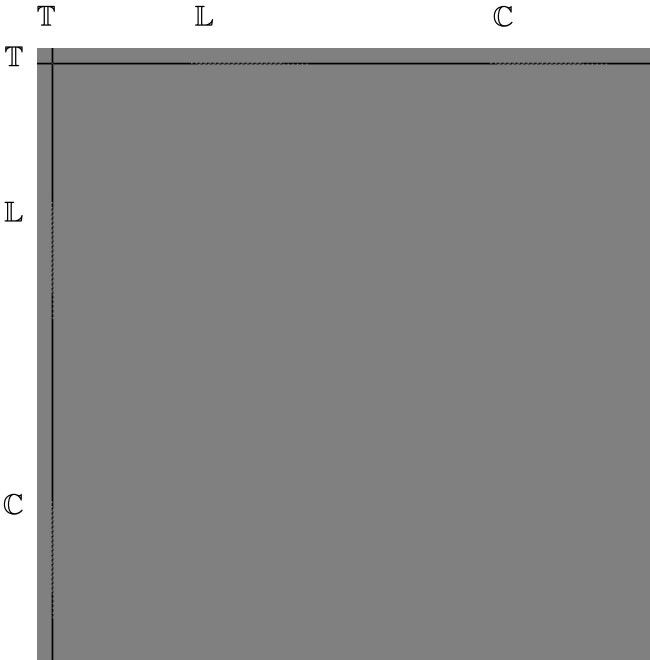

*Figure 8.* Representation of $M^{\text{har}}$ when running FLUX Kontext as a baseline on a sample from LoMOE-Bench. We use the color black to indicate areas which are blocked from attending each other and the color grey to indicate areas where attention is allowed.

## D. Further Details on the Scores

For multi-instance text content editing in the infographics benchmark datasets, we use the following scores:

- We compute the average Character Error Rate (**CER**) obtained by PaddleOCR over all edited text boxes. This score directly quantifies the correctness of the rendered text with respect to the editing prompt, thereby evaluating adherence to the requested textual content. Because of the programmatically-generated nature of the Crello dataset, we generate ground-truth renderings of images containing the target text and compute CER on these. We then subtract this ground-truth CER from the one computed on the samples generated by the evaluated methods and refer to the score thus obtained as $\Delta$**CER**, which we only report for Crello, as it is the only programmatically generated dataset in our benchmark.

- We report the **MAE** and **MSE** differences between each edited infographic and its reference image, restricted to pixels outside the specified text-box regions. These metrics, dubbed $\textbf{MAE}_\textbf{B}$ and $\textbf{MSE}_\textbf{B}$, penalize unintended changes to the original image.

- As customary for assessing the realism of the edited images compared to the original distribution, we report the Fréchet Inception Distance (**FID**) (Heusel et al., 2017) between the edited infographics and the corresponding reference images.

- We introduce **Attempt Rate** in order to evaluate the models' ability to attempt to perform edits instead of ignoring prompts. This is quantified as the percentage of target bounding boxes where the Mean Absolute Error (MAE) between the original and the edited image (both images as RGB, with each channel represented by an integer in the range $[0, 255]$) crop exceeds a fixed threshold (set to $10.0$). This score effectively serves as a "sanity check" to distinguish between failed edits (where the model modifies the region but produces poor quality) and ignored instructions (where the model simply reconstructs the original content). As with many heuristic approaches, we acknowledge that this score has some limitations, but empirically serves its purpose to distinguish areas where significant edits were applied and those in which they were not.

  In Figure 9, we report a sensitivity analysis of this score, obtained by varying the threshold on the MAE between the original and the edited image. As we can observe, the models' rankings are quite stable (especially in Crello Edit and InfoEdit, where the number of edits per image is much higher and, therefore, more statistically significant).

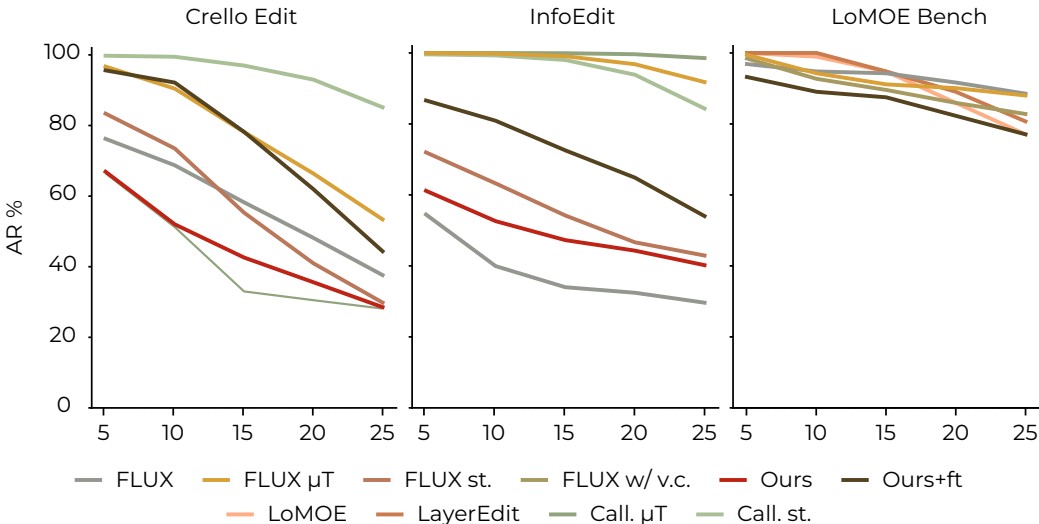

*Figure 9.* AR% sensitivity analysis with varying MAE thresholds on Crello Edit, InfoEdit, and LoMOE Bench.

For LoMOE-Bench (Chakrabarty et al., 2024), we use the scores proposed by its authors, *i.e.,*:

- **Target CLIP Score (Tgt C)** (Hessel et al., 2021). To evaluate the semantic fidelity of the edits, we measure the alignment between the edited image and the target text prompt. Following the LoMOE benchmark protocol, this is computed as the cosine similarity between their feature embeddings extracted by a pre-trained CLIP ViT-B/16 model. A higher score indicates visual content accurately reflecting the requested text description.

- **Structural Distance (SD)**. To assess the preservation of the scene's geometric layout, we compute the Structural Distance between the **original input image** and the **edited output image**. Following the official implementation of LoMOE-Bench (Chakrabarty et al., 2024), this is calculated as the Mean Squared Error (MSE) between the self-similarity maps of the keys extracted from the last layer of a DINO-trained ViT-B/8 model. This metric effectively captures high-level structural correspondence between the source and the edit, ensuring the preservation of object poses and scene composition.

- **Image Reward (IR)** (Xu et al., 2023) and **Human Preference Score (HPS)** (Wu et al., 2023). Following the evaluation protocol proposed in LoMOE-Bench (Chakrabarty et al., 2024), we employ both IR and HPS to quantify perceptual quality. Unlike standard metrics that focus on semantic alignment or pixel-level fidelity, these models are trained on human preference data, serving as robust proxies for visual realism and aesthetic appeal in editing tasks.

- **Background LPIPS (LPIPS_B)** (Zhang et al., 2018) and **Background SSIM (SSIM_B)** (Wang et al., 2004). To verify that the editing process remains localized and does not introduce unintended artifacts in the surrounding scene, we compute LPIPS and SSIM exclusively on the background regions (masked using the inverted edit mask). Background LPIPS utilizes deep features (AlexNet) to measure perceptual deviations, while Background SSIM assesses pixel-level structural consistency between the original and edited images.

Moreover, we use **FID** and **Attempt Rate (AR)**, and introduce the **Localized CLIP Score (Loc C)** To compute it, we isolate each target object by masking the surrounding context with a white background and extracting a square crop centered on its bounding box (scaled by a factor of 1.2). We then measure the cosine similarity between the CLIP embedding of this isolated view and the corresponding local editing prompt, averaging the scores across all instances.

## E. Extended Ablation Studies

We report the full set of computed scores on the ablation study on prompt encoding and application of instance-disentangled attention in Table 7, and on the ablation study on layer scheduling strategy in Table 6.

*Table 6.* Extended ablation analysis on the layer scheduling strategy.

| $L_{early}$ | $L_{mid}$ | $L_{late}$ | Tgt C↑ | SD↓ | IR↑ | HPS↑ | LPIPS$_B$ ↓ | SSIM$_B$ ↑ | Loc C↑ | AR [%]↑ |
|---|---|---|---|---|---|---|---|---|---|---|
| $M^{dis}$ | $M^{dis}$ | $M^{dis}$ | 25.51 | 0.068 | 0.566 | 0.276 | 0.108 | 0.910 | 29.14 | 94.27 |
| $M^{dis}$ | $M^{har}$ | $M^{har}$ | 25.01 | 0.069 | 0.145 | 0.270 | 0.099 | **0.928** | 28.53 | 82.29 |
| $M^{dis}$ | $M^{dis}$ | $M^{har}$ | 25.63 | 0.070 | 0.598 | 0.275 | 0.100 | 0.927 | 29.21 | 91.15 |
| $M^{dis}$ | $M^{har}$ | $M^{dis}$ | 25.03 | 0.076 | 0.161 | 0.272 | 0.097 | 0.927 | 28.62 | 80.21 |
| $M^{har}$ | $M^{har}$ | $M^{dis}$ | 25.11 | 0.073 | 0.205 | 0.272 | 0.101 | 0.922 | 28.63 | 83.33 |
| $M^{har}$ | $M^{dis}$ | $M^{dis}$ | 25.59 | 0.072 | **0.602** | 0.275 | 0.103 | 0.916 | 29.23 | 93.23 |
| $M^{har}$ | $M^{dis}$ | $M^{har}$ | **25.67** | **0.064** | 0.278 | **0.657** | **0.091** | 0.926 | **29.26** | 92.19 |
| $M^{har}$ | $M^{har}$ | $M^{har}$ | 25.05 | 0.073 | 0.188 | 0.270 | 0.103 | 0.924 | 28.54 | 80.21 |

*Table 7.* Extended ablation analysis on prompt encoding and application of instance-disentangled attention.

| Efficient prompt enc. | IDAttn | Tgt C↑ | SD↓ | IR↑ | HPS↑ | LPIPS$_B$ ↓ | SSIM$_B$ ↑ | Loc C↑ | AR$_\%$ ↑ |
|---|---|---|---|---|---|---|---|---|---|
| | | 25.22 | 0.080 | 0.270 | 0.247 | 0.126 | 0.908 | 27.93 | 86.46 |
| ✓ | | 25.16 | 0.079 | 0.270 | 0.287 | 0.129 | 0.903 | 27.90 | 86.46 |
| | ✓ | **25.67** | **0.064** | **0.278** | **0.657** | **0.091** | **0.926** | **29.26** | 92.19 |
| ✓ | ✓ | 25.60 | 0.073 | 0.277 | 0.574 | 0.099 | 0.919 | 29.08 | 89.06 |

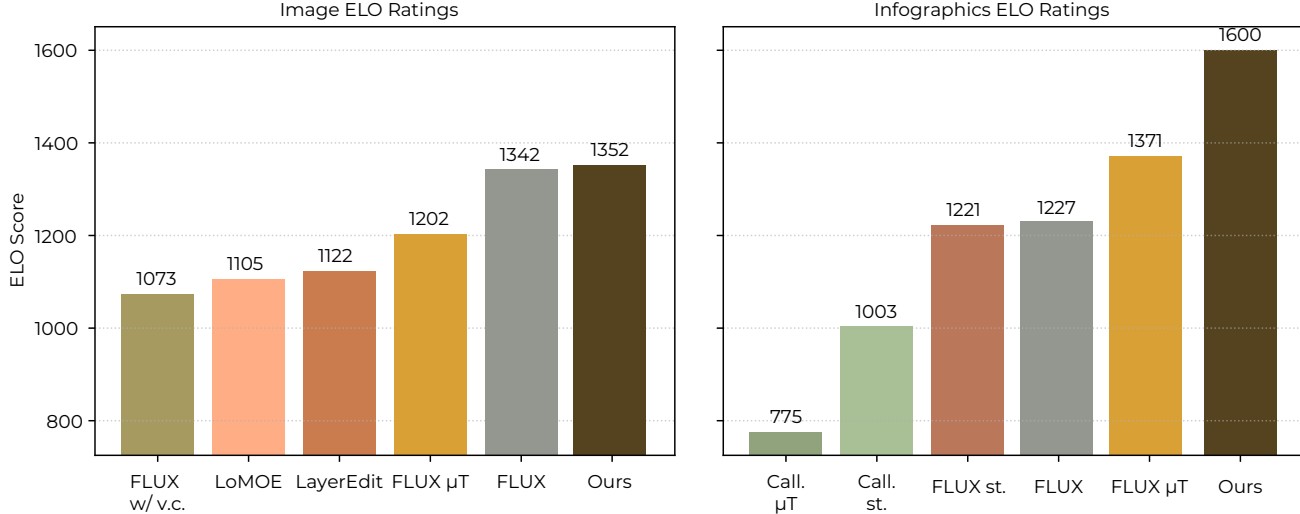

*Figure 10.* Elo score of LLM-as-a-judge on the image benchmark (LOMOE) and the Infographics (Crello and Infodet). We use the abbreviation *Call.* for the Calligrapher baselines.

## F. Extended LLM-as-a-judge Evaluation

We repeat the same experiments performed on the LLM-as-a-judge setting based on Gemini 3 Flash described in Section 5 on an extended set of questions including all binary comparisons of 6 models on 37 images from LoMOE-Bench and 119 images from the infographic datasets and we report the results in Figure 10. In this extended validation our model obtains the highest score, particularly on infographics and we find out the following: (i) on LoMOE-Bench LoMOE underperforms when compared to the scores proposed in (Chakrabarty et al., 2024) and reported in Table 3: the Elo score computed from the LLM-as-a-judge evaluation only ranks it higher than the FLUX Kontext baseline with visual cues, whereas the end-to-end FLUX Kontext baseline gets ranked as second-best, with a score that is close to our method, but significantly higher than all others; (ii) on infographics, Calligrapher in particular severely underperforms when compared to the other models, and our method leads by a significant margin over all competitors and baselines.

Especially on LoMOE-Bench, this underlines a lack of reliable scores to measure this task in a consistent way: on that dataset, LoMOE appears from the scores to be among the best models. The generation quality, though, is low, so the

| Source | Edit Instances | LoMOE | LayerEdit | Ours |

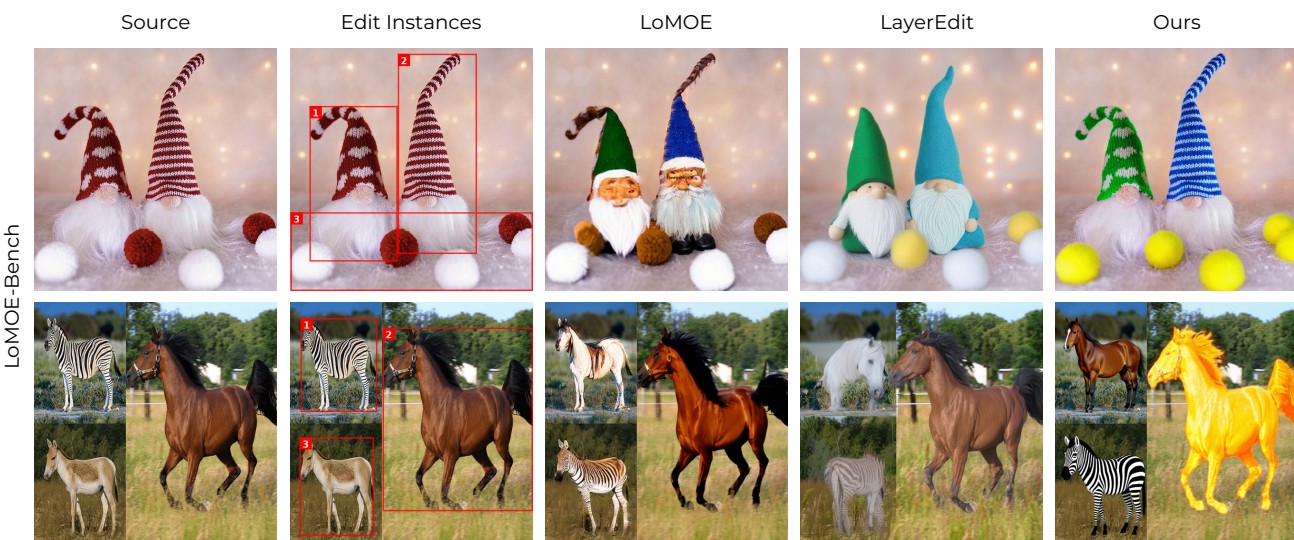

*Figure 11.* Further qualitative comparison on **LoMOE-Bench**.

LLM-as-a-judge model prefers FLUX-based models, which generate images that look much better. This is qualitatively evident by looking at samples like those in Figure 11, but is not captured by CLIP score, for example. Moreover, in Figure 12, we expand this analysis by showing scores computed on individual samples obtained on LOMOE-Bench with our model, LOMOE, and LayerEdit. All samples show how both CLIP score and especially SSIM penalize our model, even though it produces a much more realistic and well-harmonized output.

*Table 8.* Quantitative evaluation on typography and style preservation. Scores are computed exclusively on the text boxes that the models attempted to edit.

| | Crello Edit | | | | InfoEdit | |
|---|---|---|---|---|---|---|
| | **HWD↓** | **SSIM↑** | **FID↓** | **# boxes** | **FID↓** | **# boxes** |
| **Calligrapher** $\mu$**T** | 0.69 | 0.30 | 41.35 | 7010 | 183.21 | 53228 |
| **Calligrapher st.** | 0.69 | 0.33 | 25.65 | 16938 | 28.90 | 52115 |
| **FLUX** | 0.43 | 0.40 | 13.10 | 11278 | 29.52 | 20702 |
| **FLUX** $\mu$**T** | 0.28 | 0.45 | 13.12 | 15148 | 109.72 | 53137 |
| **FLUX st.** | 0.39 | 0.41 | 11.62 | 11641 | 17.25 | 33673 |
| **Ours** | 0.37 | 0.39 | 13.56 | 8422 | 21.28 | 25020 |
| **Ours + ft** | **0.20** | **0.56** | **7.82** | 15287 | **7.10** | 41030 |

## G. Extended Evaluation on Typography and Style Preservation

To provide a more reproducible and direct evaluation than qualitative comparisons, we investigate a set of scores that capture typography, style, and visual coherence preservation. note that identifying a unified score for these aspects is somewhat tricky. In this regard, we consider the scores used in the Styled Text Image Generation literature (Pippi et al., 2025; Zaccagnino et al., 2026) and compute them on the text boxes that the models are requested to edit. Specifically, we consider the FID, the task-specific Handwriting distance (HWD) (Pippi et al., 2023), and the Structural Similarity Index (SSIM). Note that HWD and SSIM are not robust to changes in text size and/or line count, which naturally occur when translating text into a different language. For these reasons, we report these scores on the paired samples from Crello Edit (recall that, for Crello Edit, the target edited images are available as ground truth). As for the FID, we report it for both infographics datasets. Moreover, we argue that these scores are significant when computed on the boxes that the model has attempted to edit. Therefore, we compute them only on such boxes, which can differ from model to model, but each still represents a statistically significant sample count. We report these quantitative results in Table 8.

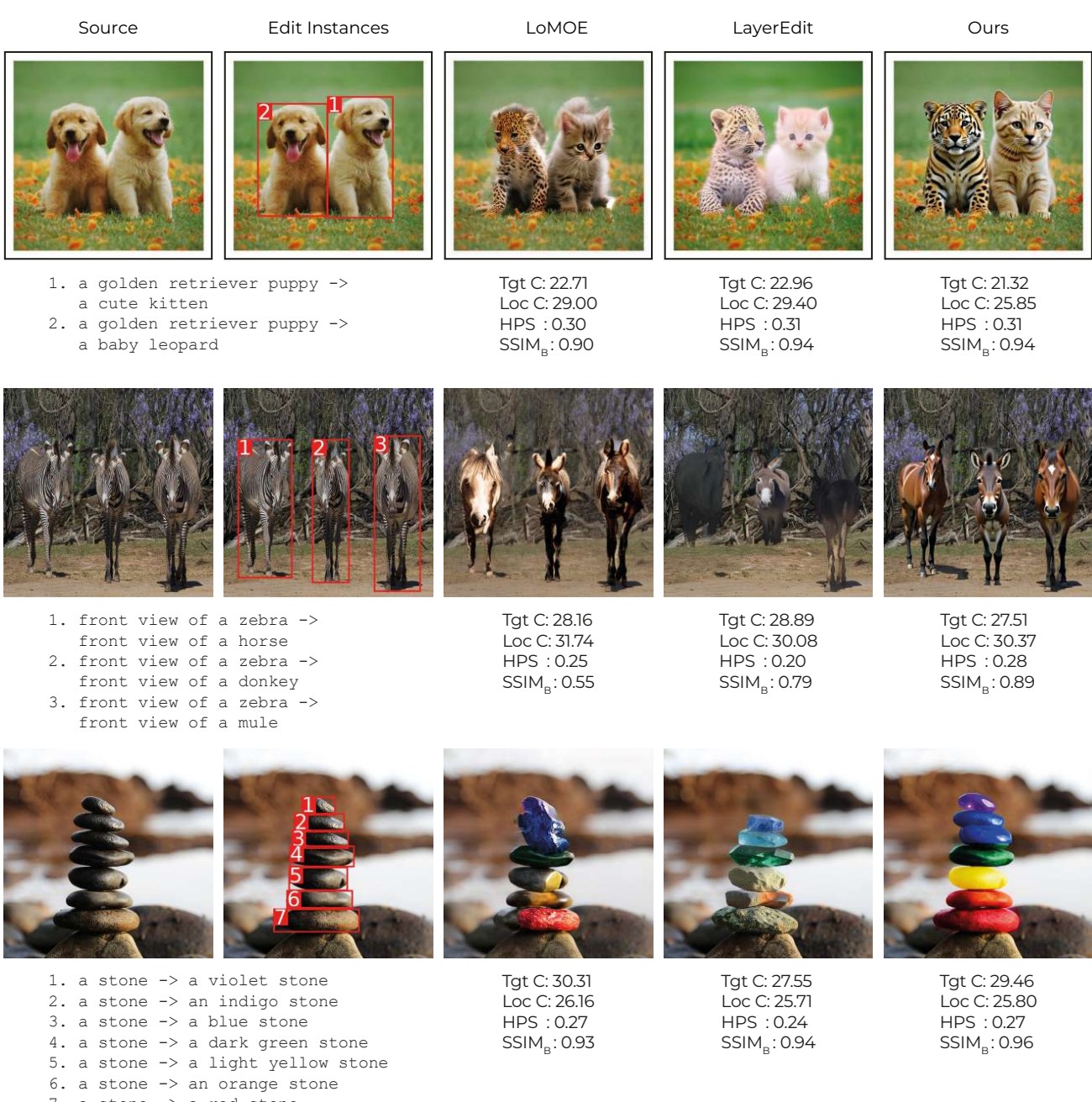

*Figure 12.* Scores computed on individual samples obtained on LOMOE-Bench with our model, LOMOE, and LayerEdit.

## H. Prompts for Qualitative Results in the Main Paper Body

The prompts for the qualitative results are presented here in the format `Instance ID. Source -> Target` which are passed to all FLUX Kontext-based models as `Replace Source with Target` instead. For other approaches, we follow their standard pipelines, including prompt generation. The promtps for the samples shown in Figure 4 are as follows.

For the Crello Edit sample:
```
1.  FOR YOU -> POUR TOI
2.  JUST -> SEULEMENT
3.  AND -> AND
```

```
4.  with care -> avec précaution
5.  WITH LOVE -> AVEC AMour
6.  MADE -> FABRIQUÉ
```

For the InfoEdit sample:
```
1.  Who's Getting The Most Sleep?.  -> Chi sta dormendo di più?
2.  Netherlands -> Paesi Bassi
3.  5.22 -> 5.22
4.  New Zealand -> Nuova Zelanda.
5.  4.00 -> 4.00
6.  France -> Francia
7.  3.19 -> 3.19
8.  Australia -> Australia
9.  1.30 -> 1.30
10.  -1.73 -> -1,73
11.  Canada -> Canada
12.  -3.16 -> -3.16
13.  United Arab Emirates -> Emirati Arabi Uniti.
14.  -6.12 -> -6.12
15.  United Kingdom -> Regno Unito.
16.  Italy -> Italia.
17.  -7.32 -> -7.32
18.  -7.88 -> -7.88
19.  United States -> Stati Uniti
20.  -8.51 -> -8.51
21.  China -> Cina
22.  -9.85 -> -9,85
23.  Switzerland -> Svizzera
24.  -12.50 -> -12,50
25.  -18.26 -> -18,26
26.  Germany -> Germania
27.  -25.88 -> -25,88
28.  Brazil -> Brasile
29.  -29.49 -> -29,49
30.  Japan -> Giappone
31.  -36.23 -> -36,23
32.  Singapore -> Singapore
33.  Latest available data:2016 -> Ultimi dati disponibili:  2016
34.  Source:  Science Advances -> Fonte:  Science Advances.
35.  statista -> Statista.
36.  Minutes above and below eight hours of sleep in selected countries -> Minuti
sopra e sotto le otto ore di sonno in paesi selezionati.
```

For the LoMOE-Bench sample:
```
1.  a gnome with a red hat -> a gnome with a green hat
2.  a gnome with a red hat -> a gnome with a blue hat
3.  white and red cotton balls -> yellow cotton balls
```

## I. Additional Qualitative Results

In this appendix we report further qualitative results for all three datasets.

The prompts for the Crello Edit qualitatives in Figure 13 are as follows.

For the sample in the first row:

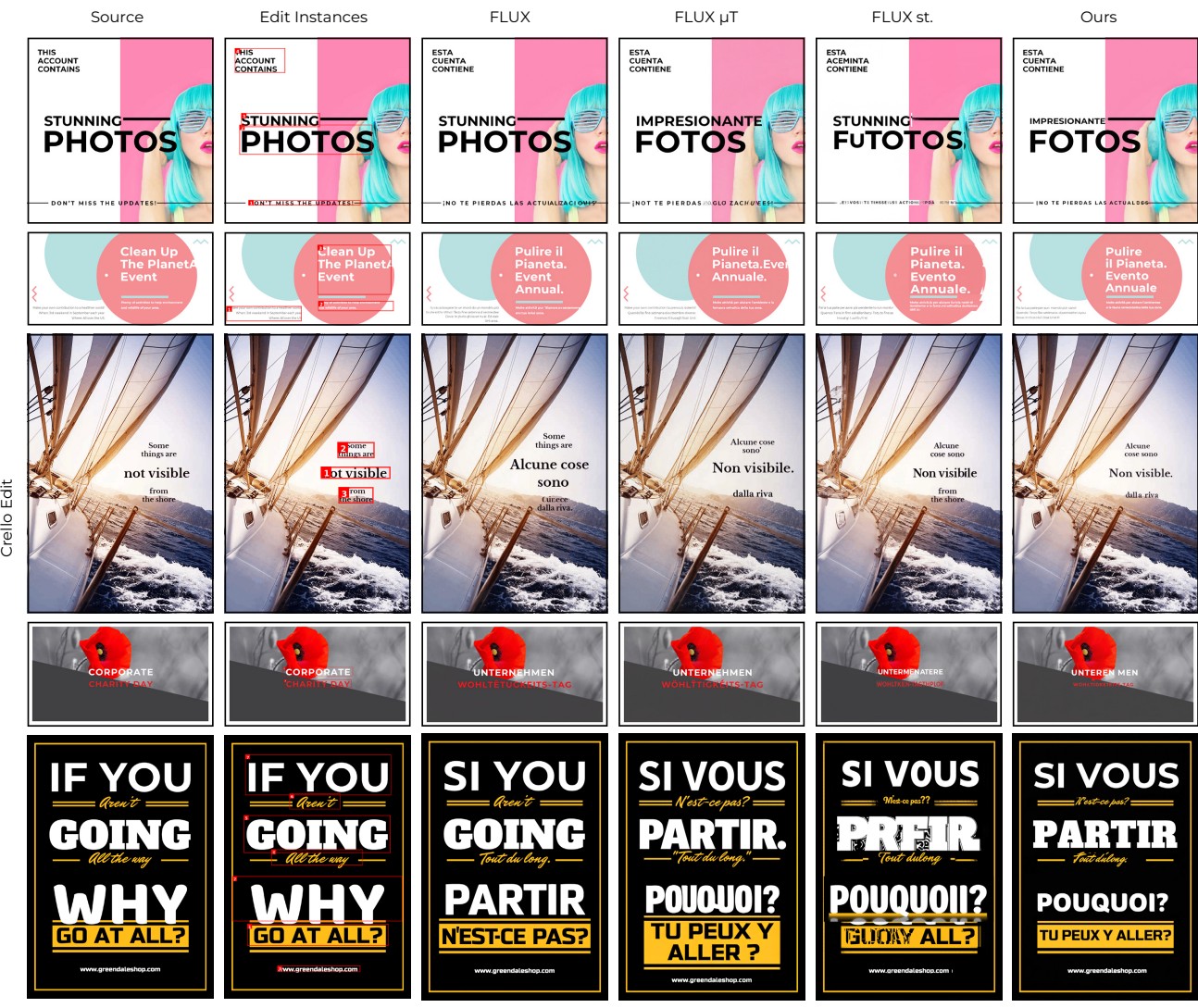

*Figure 13.* Qualitative results on the **Crello Edit** dataset.

1. DON'T MISS THE UPDATES! -> NE MANQUEZ PAS LES MISES À JOUR !
2. PHOTOS -> PHOTOS
3. STUNNING -> ÉTONNANT
4. THIS ACCOUNT CONTAINS -> CE COMPTE CONTIENT

For the sample in the second row:
1. not visible -> Non visibile.
2. Some things are -> Alcune cose sono
3. from the shore -> dalla riva

For the sample in the third row:
1. GO AT ALL? -> TU PEUX Y ALLER ?
2. WHY -> POUQUOI ?
3. www.greendaleshop.com -> www.greendaleshop.com
4. All the way -> Tout du long.
5. GOING -> PARTIR.
6. Aren't -> N'est-ce pas ?
7. IF YOU -> SI VOUS

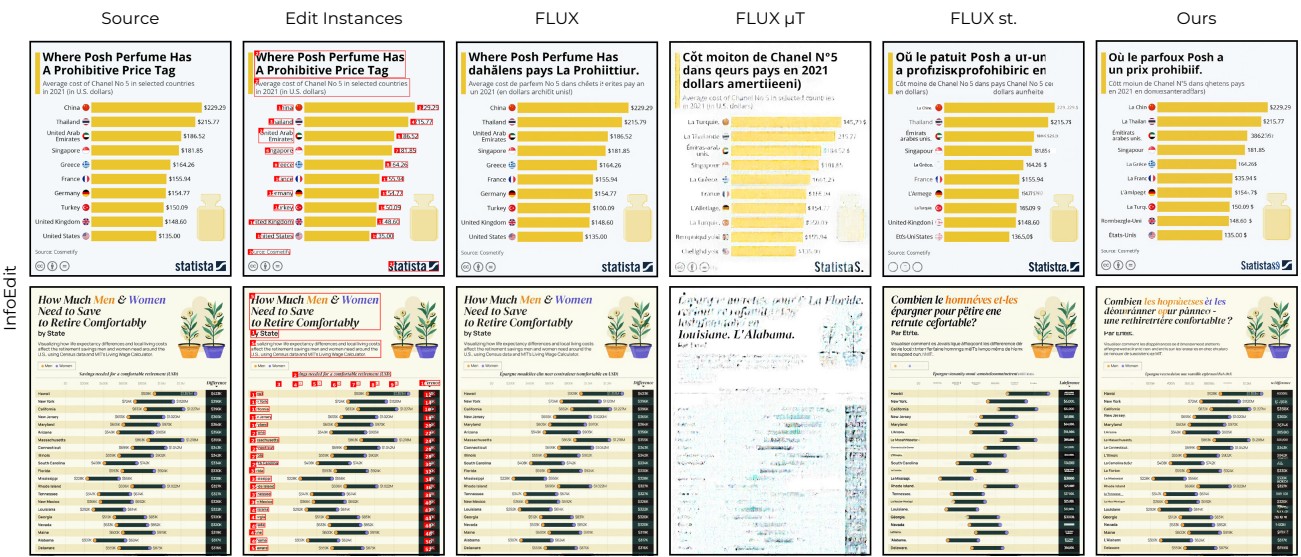

*Figure 14.* Qualitative results on the **InfoEdit** dataset.

The prompts for the InfoEdit qualitatives in Figure 14 are as follows.

For the sample in the first row:
```
1.   China -> La Chine.
2.   $229.29 -> 229,29 $
3.   Thailand -> La Thaïlande
4.   $215.77 -> 215,77 $
5.   $186.52 -> 186,52 $
6.   Singapore -> Singapour
7.   $181.85 -> 181,85 $
8.   Greece -> La Grèce.
9.   $164.26 -> 164,26 $
10.  France -> La France.
11.  $155.94 -> 155,94 $
12.  Germany -> L'Allemagne.
13.  $154.77 -> 154,77 $
14.  Turkey -> La Turquie.
15.  $150.09 -> 150,09 $
16.  United Kingdom -> Royaume-Uni
17.  $148.60 -> 148,60 $
18.  United States -> États-Unis.
19.  $135.00 -> 135,00 $
20.  Source:Cosmetify -> Source :  Cosmetify.
21.  statistaS -> StatistaS.
22.  Where Posh Perfume Has A Prohibitive Price Tag -> Où le parfum Posh a un
prix prohibitif.
23.  Average cost of Chanel No 5 in selected countries in 2021 (in U.S. dollars)
-> Coût moyen de Chanel N°5 dans certains pays en 2021 (en dollars américains)
24.  United Arab Emirates -> Émirats arabes unis.
```

For the sample in the second row:
```
1.   by State -> Par État.
2.   Savings needed for a comfortable retirement (USD) -> Épargne nécessaire pour
une retraite confortable (en USD)
```

3.  $0 -> 0 $
4.  $200K -> 200 000 dollars.
5.  $400K -> 400 000 dollars.
6.  $600K -> 600 000 dollars.
7.  $800K -> 800 000 dollars.
8.  $1.0M -> 1,0 million de dollars.
9.  $1.2M -> 1,2 million de dollars.
10.  Difference -> La différence.
11.  Hawaii -> Hawaï.
12.  $423K -> 423 000 dollars.
13.  New York -> New York.
14.  $396K -> 396 000 dollars.
15.  California -> Californie
16.  $396K -> 396 000 dollars.
17.  New Jersey -> New Jersey.
18.  $365K -> 365 000 dollars.
19.  Maryland -> Maryland
20.  $364K -> 364 000 dollars.
21.  Arizona -> L'Arizona.
22.  $356K -> 356 000 dollars.
23.  Massachusetts -> Le Massachusetts.
24.  $355K -> 355 000 dollars.
25.  Connecticut -> Le Connecticut.
26.  $343K -> 343 000 dollars.
27.  Illinois -> L'Illinois.
28.  $342K -> 342 000 dollars.
29.  South Carolina -> La Caroline du Sud.
30.  $334K -> 334 000 dollars.
31.  Florida -> La Floride.
32.  $330K -> 330 000 dollars.
33.  Mississippi -> Le Mississippi.
34.  $328K -> 328 000 dollars.
35.  Rhode Island -> Rhode Island.
36.  $327K -> 327 000 dollars.
37.  Tennessee -> Le Tennessee.
38.  $327K -> 327 000 dollars.
39.  New Mexico -> Le Nouveau-Mexique.
40.  $326K -> 326 000 dollars.
41.  Louisiana -> Louisiane.
42.  $322K -> 322 000 dollars.
43.  Georgia -> Géorgie.
44.  $320K -> 320 000 dollars.
45.  Nevada -> Nevada.
46.  $320K -> 320 000 dollars.
47.  Maine -> Le Maine.
48.  $319K -> 319 000 dollars.
49.  Alabama -> L'Alabama.
50.  $317K -> 317 000 dollars.
51.  Delaware -> Delaware.
52.  $316K -> 316 000 dollars.
53.  How Much Men & Women Need to Save to Retire Comfortably -> Combien les hommes et les femmes doivent-ils épargner pour prendre une retraite confortable
54.  Visualizing how life expectancy differences and local living costs affect the retirement savings men and women need around the U.S., using Census data and

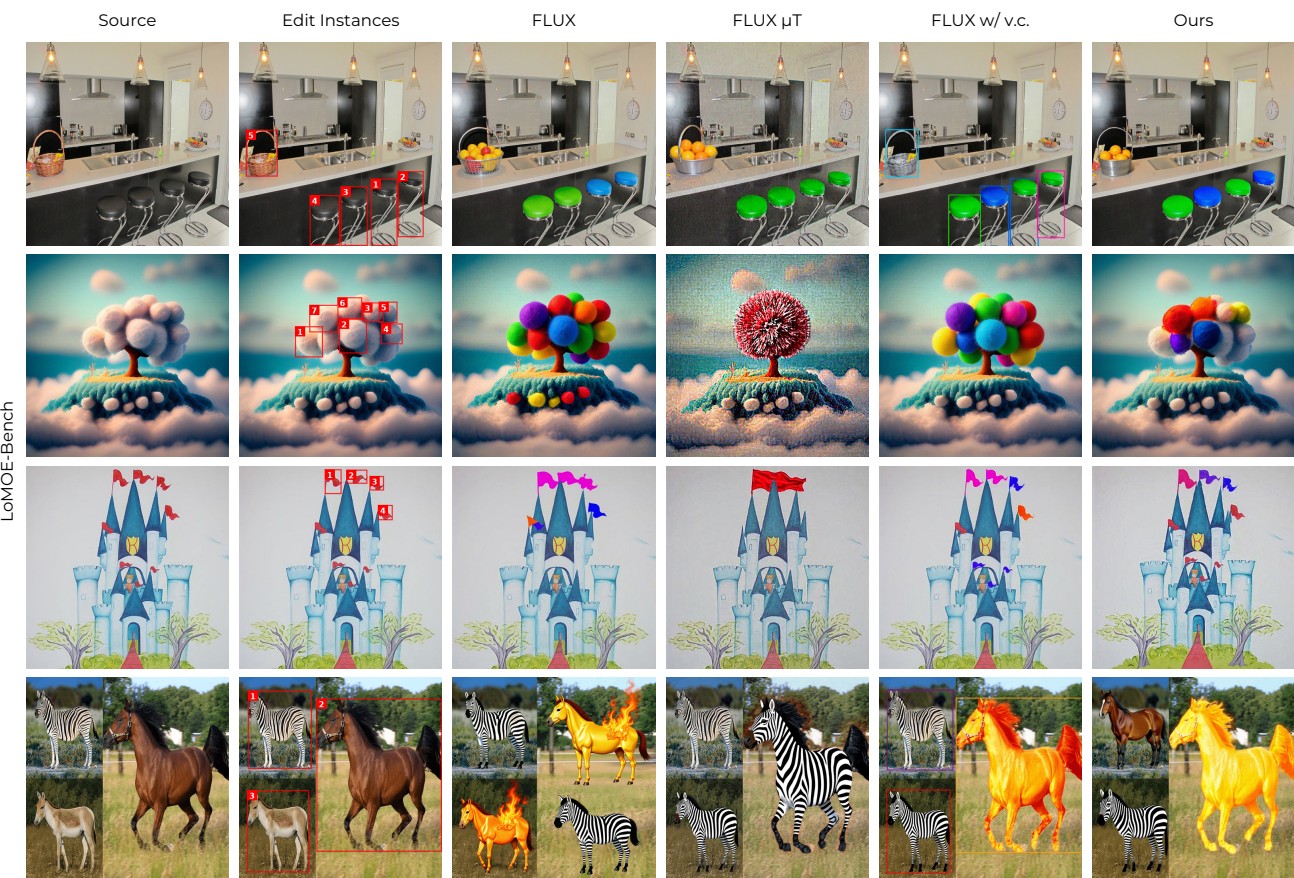

*Figure 15.* Qualitative results on the **LoMOE-Bench**.

```
MiT's Living Wage Calculator.  -> Visualiser comment les différences d'espèrance
de vie et les coûts de la vie locaux affectent l'èpargne-retraite dont les hommes
et les femmes ont besoin aux États-Unis, en utilisant les données du recensement
et le calculateur de salaire de subsistence de MiT.
```

The prompts used to generate the LoMOE-Bench qualitative results in Figure 15 are as follows.

For the sample in the first row:
```
1.  a barstool with a black cushion -> a barstool with a green cushion
2.  a barstool with a black cushion -> a barstool with a blue cushion
3.  a barstool with a black cushion -> a barstool with a blue cushion
4.  a barstool with a black cushion -> a barstool with a green cushion
5.  a wooden basket -> a steel basket with fruits
```

For the sample in the second row:
```
1.  a cotton ball -> a violet coloured cotton ball
2.  a cotton ball -> an indigo coloured cotton ball
3.  a cotton ball -> a blue coloured cotton ball
4.  a cotton ball -> a dark green coloured cotton ball
5.  a cotton ball -> a light yellow coloured cotton ball
6.  a cotton ball -> an orange coloured cotton ball
7.  a cotton ball -> a red coloured cotton ball
```

For the sample in the third row:
```
1.   a red flag -> a pink flag
2.   a red flag -> a purple flag
3.   a red flag -> a blue flag
4.   a red flag -> an orange flag
```

For the sample in the fourth row:
```
1.   a zebra -> a horse
2.   a horse -> a golden fire horse
3.   a mule -> a zebra
```

