# OpenReview forum: "Shifting the Breaking Point of Flow Matching for Multi-Instance Editing"
_ICML.cc/2026/Conference — ICML 2026 regular_

### Official Review · Reviewer_5UCs · 2026-03-10

**Soundness:** 3
**Presentation:** 4
**Significance:** 3
**Originality:** 3
**Overall Recommendation:** 3
**Confidence:** 4

**Summary:**

This paper explores how to achieve efficient and disentangled multi-instance image editing. In this study, the author intends to investigate how to prevent joint attention mechanisms in models like FLUX from entangling unrelated semantic concepts when multiple local editing instructions are provided simultaneously.

To address this, the paper proposes an architectural intervention method called Instance-Disentangled Attention (IDAttn). This method logically partitions the joint attention matrix of MMDiT during inference and introduces two masking mechanisms: a "Disentanglement Regime" in the middle layers to isolate image-text interactions of different instances, and a "Harmonization Regime" in the early and late layers to ensure global visual coherence. Additionally, the authors propose an efficient multi-prompt independent encoding strategy to reduce the computational overhead brought by long sequences. To better evaluate multi-instance editing capabilities, the authors also introduce a new infographic editing benchmark, characterized by dense text, small editing regions, and a large number of instances, providing valuable new data for this task.

**Compliance With Llm Reviewing Policy:**

Affirmed.

**Final Justification:**

I keep my original weak-reject stance because of the incremental contribution.

**Key Questions For Authors:**

Q1. Overlapping Regions:
In Equation (4), if an image token belongs to both P1 and P2 (i.e., the two bounding boxes overlap), it will attend to both T1 and T2. If the semantics of T1 and T2 are contradictory, what kind of output will the model generate? Is there a mechanism to assign priority?

Q2. Sensitivity of Layer Boundaries:
In practice, how are L_early , L_mid, and L_late  specifically divided? Does this division ratio need to be adjusted between LoMOE and InfoEdit?

Q3. Side-effects of Fine-tuning:
As shown in Table 4, why does "Ours + ft" perform worse than the pure-inference "Ours" in terms of background preservation? Does this imply that LoRA fine-tuning compromises the original image reconstruction capability of the FLUX model? Could the authors provide some intuitive visual examples to demonstrate this degradation in background preservation?

**Limitations:**

No dedicated section in the main text.

The authors do not include an independent Limitations section in the main text (only a very broad sentence in the Impact Statement in the appendix). It is strongly recommended to add a dedicated Limitations discussion in the final version. For example, the method seems to rely heavily on precise bounding boxes provided by users or external OCR systems. If the bounding boxes are inaccurate, the hard truncation of attention masks might lead to noticeable artifacts or editing failures

**Strengths And Weaknesses:**

1. Concise and effective design of IDAttn: By utilizing the functional differences of different Transformer layers to apply distinct attention masks, the proposed method ensures the accuracy of local editing while maintaining the global consistency of the image.
2. New Benchmark: Introducing Infographics as a testbed for multi-instance editing is a highly insightful contribution. It significantly expands the research boundaries in this field and provides valuable data assets for the community.
3. Presentation: The figures and visualizations in the paper are exceptionally well-crafted, greatly lowering the barrier to understanding for readers.

Weaknesses:

1. Incremental Technical Contribution: The overall idea of the paper is very clear, but from the perspective of technical novelty, applying attention masking to Transformers to prevent feature leakage is a well-established practice in the field of diffusion models. The main contribution of this paper is adapting this classic paradigm to newer MMDiT architectures like FLUX, combined with heuristic layer scheduling. While this engineering adaptation performs well in practical applications (such as infographic editing), its originality and technical depth in core methodology appear somewhat thin for a top-tier conference paper.

2.	Layer Scheduling Heuristics: Although Tables 1 and 6 provide ablation studies on layer scheduling, the specific division boundaries of LearlyL_{early}Learly​, LmidL_{mid}Lmid​, and LlateL_{late}Llate seem heuristic. The paper does not explicitly state the exact cut-offs or analyze the sensitivity of these hyperparameters to different types of images.

---

> ### Author Rebuttal · Authors · 2026-03-31
>
> ### W1
> We agree that altering the attention of diffusion models has been explored for controlling their behavior for generation and, to a much lesser extent, generative editing. However, in this work, we tackle a challenging variant of generative editing that entails performing edits on a large number of instances concurrently (as evaluated via our benchmark), which, to the best of our knowledge, is unexplored in the literature. We argue that achieving such capability is not simply a matter of scaling existing approaches via engineering optimizations, but requires designing a novel principled masking strategy to ensure both attribute disentanglement between the edited instances and overall coherence within the final output.
>
> ### Q1
>
> Our approach does not entail any explicit priority mechanism in applying the edits, which are performed concurrently. Adherence to the prompts, even when conflicting, relative to areas whose bounding boxes overlap, is ensured by the fact that our operator does not hinder the localization capabilities of the backbone model. In such cases, IDAttn allows for more intense interactions within smaller instances, due to the fact that the softmax computed over a smaller region will be less diluted by attention performed on background pixels when compared to larger regions, thus creating an implicit focus on the editing instruction which refers to the smaller box for the area that is in common between the two, which is usually a beneficial sideproduct. An example of this behaviour is reported [in the images at this link](https://imgur.com/a/0VbKG1z),  especially the one with the giraffes: prompt 3 (asking to change the color to yellow and pink) insists on the same area of prompt 4 (asking to change the color to white and green) and our approach correctly applies the edits.
>
> ### W2/Q2
>
> Sorry, we forgot to explicitly state this implementation detail. We adopted the same setting proposed in [B], using layer 10 as the cut-off from $L_\text{early}$ to $L_\text{mid}$ and layer 47 as the cut-off from $L_\text{mid}$ to $L_\text{late}$, and used the same setting across all experiments without any tuning. We argue that these cut-offs are appropriate to ensure that the majority of layers run in the disentanglement regime, and only the first and last 10 layers run in the harmonization regime to ensure the build-up of a coherent latent representation of the image within the MMDiT, and to harmonize the content to avoid artifacts due to the mask cut-off.
>
> ### Q3
>
> We are afraid there has been some confusion: as reported in Table 4, the “Ours+ft” variant always outperforms the “Ours” variant in terms of background preservation as measured by MSE and MAE computed on the background pixels (61.35/90.86 vs 61.72/91.58 on Crello Edit, and 3.22/13.41 vs 3.41/15.33 on InfoEdit). As for the other scores, the two variants lie in the same ballpark. From this, it can be concluded that LoRA finetuning gives the model a better understanding of text-heavy editing prompts, improving its ability to follow them without affecting its other capabilities.
>
> ### Limitations
>
> We agree with the reviewer that a dedicated limitations discussion paragraph would be interesting and helpful. We concur with the reviewer that a limitation of our approach for multi-instance editing is that, like previous approaches (e.g., LoMOE and LayerEdit), it relies on external inputs to localize instances. Nonetheless, devising an end-to-end approach that can localize and apply the prompt-specified edits is a challenging goal beyond the scope of this work, which focuses on the editing component (which remains challenging even when perfect localization is provided). This is certainly an interesting future direction, and we argue that it could reasonably be tackled, e.g., with agentic pipelines orchestrating state-of-the-art promptable detection or segmentation approaches for natural images, and text detection and OCR approaches for infographics, in addition to MMDiT-based editing models equipped with strategies for multi-instance concurrent editing such as ours.
>
> Editing failures can occur when highly inaccurate or completely wrong boxes are provided. However, our approach is robust to imprecise boxes (too tight or too broad, as those in [this image](https://imgur.com/a/GSpXOYb) and in [this image](https://imgur.com/a/0VbKG1z)) that can still reasonably identify the instance of interest. As for possible artifacts caused by hard truncation when localizing using bounding boxes, these are prevented by the harmonization biasing regimes (e.g., in the bottom-right image of the llama qualitative, the crochet and wooden textures are blended rather than abruptly interrupted).
>
> Such discussion and qualitative analysis will be reported in the revised version of the manuscript.
>
>
> ---
>
> [B]: Zhou, Dewei, et al. "DreamRenderer: Taming multi-instance attribute control in large-scale text-to-image models." ICCV. 2025.

---

> > ### Author Rebuttal · Reviewer_5UCs · 2026-04-04
> >
> > The rebuttal adequately clarifies several implementation details I asked about, including the fixed layer boundaries, the behavior under overlap, the AR sensitivity analysis, and the intended limitations discussion. These responses improve confidence in the soundness and presentation.
> >
> > However, they do not materially change my main reservation: the technical contribution still feels incremental, consisting mainly of adapting familiar attention-masking ideas to MMDiT-based editing with a heuristic schedule, while the stronger empirical story is driven more by engineering and benchmark design than by methodological depth. The new infographic benchmark is valuable, and the paper has practical merit, but for the ICML bar I still find the originality and core technical novelty somewhat limited. I therefore keep my original weak-reject stance.

---

> > > ### Author Response · Authors · 2026-04-07
> > >
> > > We thank the Reviewer for their time and suggestions. We respectfully disagree with their assessment that “the stronger empirical story is driven more by engineering and benchmark design than by methodological depth” because the results are consistently strong across multiple benchmarks (not only our proposed one) on a task for which this approach (inference-time joint attention biasing) is essentially unexplored in the literature, and whose details must therefore be designed in a principled way in order for it to be effective.
> > >
> > > We understand that “quantifying” the originality and novelty of a submission is not trivial and we value the Reviewer’s point. Our self-assessment was guided by the  Reviewer Instructions for ICML, which we report here for convenience (bolding is present in the original formatting):
> > >
> > > > Does the work provide new insights, deepen understanding, or highlight important properties of existing methods? Does the work introduce new tasks, methods, theory, data, or perspectives that advance the field in some dimensions? Does this work offer a novel combination of existing techniques, and is the reasoning behind this combination well-articulated? Are the contributions clearly distinguished from closely related literature, and is the novelty well justified? **As the questions above indicate, originality does not necessarily require introducing an entirely new method. Rather, a work that provides novel insights by evaluating existing methods, or demonstrates improved understanding is also equally valuable.**
> > >
> > > In light of this guideline and the contributions of our paper, we respectfully would like to ask the Reviewer to reconsider their evaluation of the originality and novelty of our work and its potential contribution to the community.

---

### Official Review · Reviewer_2P5i · 2026-03-10

**Soundness:** 3
**Presentation:** 4
**Significance:** 4
**Originality:** 3
**Overall Recommendation:** 5
**Confidence:** 3

**Summary:**

This paper tackles "attribute leakage" in multi-instance image editing for Flow Matching models via an efficient, single-pass approach. The core innovation is Instance-Disentangled Attention (IDAttn), which uses layer-wise masking to isolate regional edits in middle layers while maintaining global coherence in early and late layers. Combined with independent multi-prompt encoding, it drastically reduces computational costs. The authors also introduce a novel Infographics Editing Benchmark, where experiments prove the method's superior ability to achieve precise, disentangled edits in dense and complex visual layouts.

**Compliance With Llm Reviewing Policy:**

Affirmed.

**Final Justification:**

I keep my original score

**Key Questions For Authors:**

1.  Since the method relies on bounding boxes, how does it handle highly irregular objects where a rectangle captures excessive background?
2.  Section 3.2 mentions that overlapping instances share tokens across partitions. In extreme cases (e.g., a small box entirely inside a larger one), does the $M^{dis}$ mask lose its effectiveness?
3. The Attempt Rate (AR%) uses a fixed MAE threshold of 10.0. How sensitive are the benchmark rankings to this specific value? A brief sensitivity analysis would strengthen this metric.

**Limitations:**

No. While the authors include an Impact Statement and briefly note the limitations of the AR% metric in Appendix D, they omit a dedicated discussion of architectural limitations in the main text (e.g., reliance on bounding boxes and failure modes during severe overlap). Adding a brief "Limitations" paragraph to the conclusion is highly recommended to provide a more balanced view of the work.

**Strengths And Weaknesses:**

Soundness: The technical foundation is solid. Partitioning the MMDiT token space via layer-wise attention masks ($M^{dis}$ for middle-layer isolation and $M^{har}$ for early/late-layer coherence) is logical and backed by thorough ablations. While the experiments are well-crafted, using a hardcoded threshold for the "Attempt Rate" metric introduces a minor empirical weakness.

Presentation: The manuscript is well-structured and highly readable. It effectively contextualizes the proposed approach against current diffusion and flow-matching baselines. Furthermore, the detailed attention mask diagrams in the appendix greatly aid reproducibility.

Significance: This work offers substantial practical value by drastically reducing computational overhead through single-pass disentangled editing. Additionally, the newly introduced Infographics Editing Benchmark provides the community with a rigorous, much-needed testbed for text-dense, layout-strict generation tasks.

Originality: Adapting attention masking to the joint attention mechanism of MMDiT-based Flow Matching models (e.g., FLUX.1) is a clever application. Combining this with an efficient multi-prompt encoding strategy showcases strong engineering originality, directly addressing a real-world bottleneck.

The primary weaknesses are the method's strict reliance on rectangular bounding boxes—which may fail on irregular shapes—and the lack of discussion regarding model behavior under severe bounding box overlap.

---

> ### Author Rebuttal · Authors · 2026-03-31
>
> ### Q1
>
> Our approach relies on additional information to localize instances (i.e., identify which visual tokens belong to which instance to be edited). This can be provided as bounding boxes, as in the experiments in the submitted paper, or with something different, such as segmentation masks. In this latter case, irregular shapes do not affect how much background is contained in the instance’s bounding box.
>
> In any case, even when using bounding boxes that include the background, the backbone model can still rely on its original localization capabilities, and the editing is correctly applied to the intended pixels (for example, please see the cotton balls within their broad bounding box in the qualitative examples [at this link](https://imgur.com/a/0VbKG1z)).
>
> ### Q2
>
> Our proposed IDAttn operator can still be effective in the cases of nested boxes mentioned by the reviewer (e.g., two giraffes in the qualitative [at this link](https://imgur.com/a/0VbKG1z)). In such cases, the backbone model can still rely on its localization capabilities, which (when they fail because of semantic equivalence between the source object in the two objects, such as in the case of the giraffes) are further enhanced by our biasing strategy, as explained further in our answer to Question 1 from Reviewer 5UCs.
>
> ### Q3
>
> We thank the reviewer for this suggestion. As a sensitivity analysis of the AR%, we considered varying the MAE threshold to 5.0, 10.0, 15.0, 20.0, and 25.0 and report the results in the following tables. It can be observed that the models’ rankings are quite stable (especially in Crello Edit and InfoEdit, where the number of edits per image is much higher and, therefore, more statistically significant). Inspired by this comment, we will report the mean and standard deviation of the AR% in the revised version.
>
> | **CRELLO EDIT** | **AR@5** | **AR@10** | **AR@15** | **AR@20** | **AR@25** | **MEAN** | **STD** |
> |---|---|---|---|---|---|---|---|
> | Calligrapher multi-turn | 66.48 | 51.21 | 33 | 30.52 | 28.14 | 41.87 | 8.88 |
> | Calligrapher stitching | 99.77 | 99.47 | 96.98 | 93 | 85.17 | 94.88 | 5.65 |
> | Kontext E2E | 76.23 | 68.72 | 58.24 | 48.18 | 37.63 | 57.80 | 12.10 |
> | Kontext multi-turn | 96.7 | 90.44 | 78.15 | 66.41 | 53.41 | 77.02 | 14.78 |
> | Kontext stitch | 83.37 | 73.49 | 55.31 | 40.98 | 29.84 | 56.20 | 17.54 |
> | Ours | 66.91 | 52 | 42.6 | 35.61 | 28.55 | 44.87 | 9.76 |
> | Ours+ft | 95.62 | 92.16 | 78.2 | 61.98 | 44.38 | 74.47 | 18.52 |
>
> | **INFOEDIT** | **AR@5** | **AR@10** | **AR@15** | **AR@20** | **AR@25** | **MEAN** | **STD** |
> |---|---|---|---|---|---|---|---|
> | Calligrapher multi-turn | 100 | 99.98 | 99.93 | 99.64 | 98.56 | 99.62 | 0.60 |
> | Calligrapher stitching | 99.61 | 99.31 | 98.02 | 93.86 | 84.34 | 95.03 | 6.18 |
> | Kontext E2E | 54.47 | 39.94 | 34 | 32.44 | 29.64 | 38.10 | 4.16 |
> | Kontext multi-turn | 99.97 | 99.81 | 99.09 | 96.83 | 91.8 | 97.50 | 3.31 |
> | Kontext stitch | 71.98 | 63.25 | 54.21 | 46.65 | 42.93 | 55.80 | 8.79 |
> | Ours | 61.16 | 52.61 | 47.27 | 44.3 | 40.21 | 49.11 | 4.72 |
> | Ours+ft | 86.61 | 80.9 | 72.56 | 64.87 | 54.04 | 71.80 | 10.23 |
>
> | **LOMOE BENCH** | **AR@5** | **AR@10** | **AR@15** | **AR@20** | **AR@25** | **MEAN** | **STD** |
> |---|---|---|---|---|---|---|---|
> | LoMOE | 100 | 98.96 | 94.79 | 85.94 | 77.08 | 91.35 | 9.71 |
> | LayerEdit | 100 | 100 | 94.79 | 89.06 | 80.73 | 92.92 | 8.17 |
> | Kontext | 96.88 | 94.79 | 94.27 | 91.67 | 88.54 | 93.23 | 3.21 |
> | Kontext w/ v.c. | 98.44 | 92.71 | 89.58 | 85.94 | 82.81 | 89.90 | 6.06 |
> | Kontext multi-turn | 99.48 | 94.27 | 91.15 | 90.1 | 88.02 | 92.60 | 4.46 |
> | Ours | 93.23 | 89.06 | 87.5 | 82.29 | 77.08 | 85.83 | 6.27 |
>
> ### Limitations
>
> We agree with the reviewer that a dedicated limitations discussion paragraph would be interesting and helpful. We concur with the reviewer that a limitation of our approach for multi-instance editing is that, like previous approaches (e.g., LoMOE and LayerEdit), it relies on external inputs to localize instances. Nonetheless, devising an end-to-end approach that can localize and apply the prompt-specified edits is a challenging goal beyond the scope of this work, which focuses on the editing component (which remains challenging even when perfect localization is provided). This is certainly an interesting future direction, and we argue that it could reasonably be tackled, e.g., with agentic pipelines orchestrating state-of-the-art promptable detection or segmentation approaches for natural images, and text detection and OCR approaches for infographics, in addition to MMDiT-based editing models equipped with strategies for multi-instance concurrent editing such as ours.
>
> As for the failure modes when there is overlap between boxes, our approach also works when the instance localization is given in the form of segmentation masks (which do not overlap - for further details on this point, please refer to the response to Q1), and it can also perform edits when the instances overlap.

---

> > ### Author Rebuttal · Reviewer_2P5i · 2026-04-03
> >
> > The authors resolved my concerns.

---

> > > ### Author Response · Authors · 2026-04-07
> > >
> > > We thank the Reviewer for their time and appreciation. We will include the suggested additions in the revised manuscript.

---

### Official Review · Reviewer_naC4 · 2026-03-12

**Soundness:** 3
**Presentation:** 3
**Significance:** 2
**Originality:** 2
**Overall Recommendation:** 4
**Confidence:** 4

**Summary:**

This paper studies multi instance image editing for flow matching and MMDiT based editors. The authors argue that when many localized instructions are applied jointly, global velocity estimation and joint attention lead to semantic entanglement across instances. To address this, the paper proposes Instance Disentangled Attention, which partitions attention with a disentanglement mask in the middle layers and a harmonization mask in the early and late layers, together with an efficient multi prompt encoding scheme. The method is evaluated on LoMOE Bench and on a new infographic text editing benchmark with Crello Edit and InfoEdit, and the paper also explores optional LoRA fine tuning on Crello for the infographic setting.

**Compliance With Llm Reviewing Policy:**

Affirmed.

**Final Justification:**

The authors addressed my concern, so I remain positive about the paper and keep my score at 4.

**Key Questions For Authors:**

- For the infographic benchmark, the authors also use finetuned version of the model. How should readers interpret the main contribution: as an inference time method, or as an architecture plus domain adaptation recipe?
- How robust is IDAttn to imperfect localization especially overlapping boxes? Since tokens can belong to multiple partitions when boxes overlap, I would like to understand whether performance degrades gracefully under noisy boxes.
- The benchmark and metrics focus strongly on text correctness through OCR based CER and the heuristic Attempt Rate score. Could you provide additional evidence that the method preserves typography, style, and overall visual coherence, beyond Elo comparisons and qualitative examples?

**Limitations:**

Yes.

**Strengths And Weaknesses:**

Strengths

- The paper targets a relevant failure mode for current FM and MMDiT editors: localized edits interfere with each other as the number of requested edits grows. This is a practically meaningful problem, and the infographic setting is a strong stress test because it contains many small editable regions.
- The core method is technically clean and easy to follow. The token partitioning, the two masking regimes, and the layerwise schedule are clearly specified, and the paper includes ablations for both layer scheduling and prompt encoding.
- The benchmark contribution is valuable. Crello Edit and InfoEdit are substantially larger and denser than LoMOE Bench, with up to 25 edits per image in Crello Edit and up to 285 in InfoEdit, which makes the evaluation setting more realistic for dense multi instance editing.
- The empirical evidence is reasonably broad: natural images, infographics, ablations, scaling with number of edits, and a user study plus LLM as a judge evaluation. In the preference evaluation, the proposed method is favored over the FLUX baselines on both LoMOE Bench and infographics.

Weaknesses

- The method appears technically sound, but the evidence only partially supports the strongest single pass multi instance editing claim on natural images. On LoMOE Bench, the method is not uniformly best: LoMOE and/or LayerEdit are stronger on Target CLIP, Local CLIP, and Attempt Rate, while the proposed method is stronger mainly on HPS and background preservation metrics.
- In the infographic setting, the strongest results rely in part on domain specific fine tuning. Without fine tuning, the proposed method has relatively low Attempt Rate on both Crello Edit and InfoEdit, and the fine tuned version improves substantially, especially on InfoEdit. This weakens the narrative that the main contribution is a broadly effective inference time method.
- Some evaluation choices have limitations that the paper itself partly acknowledges. Attempt Rate is heuristic and can overestimate success for stitching baselines, and the infographic benchmark relies heavily on OCR and translated text, so the metrics emphasize text replacement success more than holistic visual quality or style consistency.

---

> ### Author Rebuttal · Authors · 2026-03-31
>
> ### W1
>
> On LoMOE Bench, LoMOE appears to be a strong competitor in terms of scores. However, it should be noted that individual numerical scores are often limited in holistically expressing the actual capabilities of generative models (as shown in [the figure at this link](https://imgur.com/a/I4VZDma)) and must be considered alongside other scores, qualitative results, and user preference. As we reported in Figure 8, and by the LLM-as-a-judge in the appendix (which correlates with user preference, see Table 5), the outputs of LoMOE and LayerEdit are of lower quality than ours.
>
> ### Q1/W2
>
> Our main contribution is an inference-time attention biasing strategy that can be applied on top of pre-trained MMDiT-based models, without additional training. This method consistently improves performance on both natural images and infographics, and is intended as a general, plug-and-play enhancement.
> We additionally explore lightweight domain adaptation by finetuning the model with our modifications, particularly in the infographic setting, which is significantly out-of-domain for most available backbones since it entails images with many small text-heavy regions. This finetuning step is optional and incurs modest computational overhead (only 0.90% of the parameters are trained), but it helps the model better interpret complex editing prompts in these regions, leading to better performance on infographics.
> Nonetheless, we point out that our general inference-time method improves the performance of the baseline end-to-end backbone on both natural images and infographics (our approach without finetuning exhibits comparable performance across most scores, including on infographic images).
> Inspired by the reviewer’s comment, we will review Section 3.4 to clarify the role of finetuning.
>
> ### Q2
>
> We observe that our proposed attention biasing mechanism is robust to imperfect localization. We argue that this is due to the fact that, even though we restrict the context in most layers to the instance-specific bounding boxes, the backbone model can still rely on its original localization capabilities within the instance’s box (as an example, please refer to the broad bounding box relative to the cotton balls in [this qualitative link](https://imgur.com/a/0VbKG1z)).
> As for cases where bounding boxes fully overlap (e.g., two giraffes in [the qualitative at this link](https://imgur.com/a/0VbKG1z)), these capabilities are enhanced by our biasing strategy coupled with the normalization intrinsic to attention (the softmax operation), as explained further in our answer to Question 1 from Reviewer 5UCs. We can expect that, in the case of nested boxes, the edit for the smaller region is performed in case of conflicts (as is the case for the aforementioned giraffes).
>
> ### W3/Q3
>
> We agree with the reviewer that a score that captures typography, style, and visual coherence preservation would be very helpful, more reproducible, and more direct than qualitative comparisons. However, identifying such a score is somewhat tricky.
> In this regard, we considered the scores used in the Styled Text Image Generation literature and computed them on the text boxes that the models are requested to edit.
> Specifically, we borrow the FID, the task-specific HWD [E], and the SSIM. However, HWD and SSIM are not robust to changes in text size and/or line count, which would naturally happen when translating text into a different language. For these reasons, we report these scores on the paired samples from Crello Edit (recall that for Crello Edit, the target edited images are available as ground truth). As for the FID, we report it for both infographic datasets. Moreover, we observe that these scores are significant when computed on the boxes that the model has attempted to edit. Therefore, we compute them only on such boxes, which can differ from model to model, but each still represents a statistically significant sample count.
>
>
> | | **Crello Edit** | | | | **InfoEdit** | | |
> |---|---|---|---|---|---|---|---|
> | | **HWD↓** | **SSIM↑** | **FID↓** | **#boxes** | **FID↓** | **#boxes** |
> | Calligrapher multi-turn | 0.69 | 0.30 | 41.35 | 7010 | 183.21 | 53228 |
> | Calligrapher stitching | 0.69 | 0.33 | 25.65 | 16938 | 28.90 | 52115 |
> | FLUX | 0.43 | 0.40 | 13.10 | 11278 | 29.52 | 20702 |
> | FLUX multi-turn | 0.28 | 0.45 | 13.12 | 15148 | 109.72 | 53137 |
> | FLUX stitching | 0.39 | 0.41 | 11.62 | 11641 | 17.25 | 33673 |
> | Ours | 0.37 | 0.39 | 13.56 | 8422 | 21.28 | 25020 |
> | Ours+ft | 0.20 | 0.56 | 7.82 | 15287 | 7.10 | 41030 |
>
> Finally, regarding overall visual coherence, we refer to Table 4 in the paper. Our method achieves the lowest global FID and the lowest background error (MAE and MSE) among the baselines, demonstrating that the aesthetic of the image is preserved.
>
> ---
>
> [E]: Pippi, Vittorio, et al. "HWD: A Novel Evaluation Score for Styled Handwritten Text Generation." BMVC. 2023

---

> > ### Author Rebuttal · Reviewer_naC4 · 2026-04-01
> >
> > The authors addressed my concern, so I remain positive about the paper and keep my score at 4.

---

> > > ### Author Response · Authors · 2026-04-07
> > >
> > > We thank the Reviewer for their time and insightful comments. We are confident that integrating them will strengthen the paper.

---

### Official Review · Reviewer_K2rb · 2026-03-13

**Soundness:** 1
**Presentation:** 2
**Significance:** 2
**Originality:** 3
**Overall Recommendation:** 4
**Confidence:** 4

**Summary:**

This paper introduces Instance-Disentangled Attention to address the issue of multi-instance image editing.  The authors believe this issue is caused by the entanglement of globally conditioned velocity fields and joint attention mechanisms. Instance-Disentangled Attention use attention mask to disentangle instances. The authors also introduce the editing benchmark based on infographics. The authors evaluate their method on natural image editing and a newly introduced infographic editing benchmark, and partially improve the performance.

**Compliance With Llm Reviewing Policy:**

Affirmed.

**Final Justification:**

The authors resolved my concerns. Therefore, I raise the rating to weak accept.

**Key Questions For Authors:**

1. Why are globally conditioned velocity fields and joint attention mechanisms the main reason that causes multi-instance image editing issues?

2. What is the performance of efficiency?

**Limitations:**

yes

**Strengths And Weaknesses:**

Strengths

1. The authors introduce noval benchmark. It could be a contribution of community if this benchamark is released.



Weaknesses

1. The observation of the multi-instance image editing issue is unclear. The authors do not provide any observation that supports this issue due to globally conditioned velocity fields and joint attention mechanisms, and this observation is the reason why they introduce Instance-Disentangled Attention. In other words, the authors do not “identify” the imitation.

2. The performance does not have significant gains. The author claims that their method is more efficient but does not provide the performance of efficiency.

---

> ### Author Rebuttal · Authors · 2026-03-31
>
> ### Q1/W1
>
> We thank the reviewer and respectfully disagree that the root cause is unidentified or unsupported in our manuscript. We identify and support this observation in Section 1 (Introduction) and Section 3 (Proposed Approach). We further validate it empirically. Specifically, we detail how standard text encoding and attention allow semantic tokens associated with one subject to attend to spatial regions corresponding to another, causing concept bleeding across boundaries.
> We find strong consensus in the literature that global velocity fields and unconstrained attention mechanisms entangle instances via three factors, all of which are active in joint attention:
> 1. Processing the text prompt as a unique sequence binds together attributes of different instances. Works like [A] and [B] demonstrate how this causes inter-instance attribute interference in text-to-image (T2I) generation.
> 2. Queries from different instances routinely attend to tokens of competing instances [C, D]. Notably, [D] empirically proves that semantic similarity between subjects causes their cross-attention queries to mix, leading directly to attribute bleeding.
> 3. Self-attention layers blend features across an image, inherently leading to information leakage (both visual and semantic) across disjoint instances [D].
>
> Because MMDiTs combine self- and cross-attention into a single joint attention operation, they inherently inherit and exacerbate these problems, as proven in [B].
>
> Finally, our results provide empirical evidence to support this observation. Our baselines (e.g., vanilla FLUX) utilize the exact globally conditioned velocity fields and unconstrained joint attention that the reviewer questions. As shown in Figure 4 and Figure 11, these baselines consistently fail at multi-instance editing, exhibiting severe attribute leakage and editing failures. By introducing Instance-Disentangled Attention, we address this leakage and preserve edits (Table 3 and Table 4). We hope this clarifies that our core observation is supported by prior literature, explicitly referenced in the text, and demonstrated by our empirical results.
>
> ### Q2/W2
>
> We direct the reviewer to Figure 3 from the paper, which shows how our approach's inference time scales with the number of edits performed on the same image. This is compared against the baseline FLUX used end-to-end and in multi-turn settings, as well as against the case of not using our proposed prompt encoding strategy. For better readability, we present the tabular form of Figure 3 (mean inference time in seconds, calculated over 5 samples on a system with an NVIDIA B100 GPU).
>
> | **# edits** | **FLUX** | **FLUX μT** | **Vanilla Prompt Enc.** | **Ours** |
> |---|---|---|---|---|
> | 1 | 14,23 | 14,22 | 27,69 | 27,54 |
> | 2 | 14,21 | 28,42 | 28,52 | 27,42 |
> | 3 | 14,22 | 42,64 | 29,32 | 27,87 |
> | 4 | 14,21 | 56,83 | 30,41 | 27,72 |
> | 5 | 14,22 | 71,02 | 31,05 | 28,03 |
> | 10 | 14,22 | 142,07 | 35,72 | 28,16 |
> | 15 | 14,21 | 213,09 | 40,22 | 28,52 |
> | 20 | 14,24 | 284,08 | 45,52 | 28,89 |
> | 25 | 14,24 | 355,07 | 50,97 | 29,46 |
> | 30 | 14,22 | 426,02 | 56,48 | 29,59 |
> | 40 | 14,22 | 568,08 | 68,48 | 30,23 |
> | 50 | 14,21 | 710,09 | 81,93 | 31,17 |
> | 75 | 14,23 | 1065,12 | 120,39 | 32,97 |
> | 100 | 14,23 | 1420,15 | 166,23 | 34,58 |
>
> From an analytical point of view, we can express the time and memory complexity of these models as follows:
>
> * FLUX:
>     - time: $\mathcal{O}({|Z|}^2)$
>     - memory: $\mathcal{O}({|Z|}^2)$
> * FLUX μTurn:
>     - time: $\mathcal{O}(N \cdot {|Z|}^2)$
>     - memory: $\mathcal{O}({|Z|}^2)$
> * Vanilla Prompt Enc.:
>     - time: $\mathcal{O}((|Z^\text{latent}| + |Z^\text{context}| + N \cdot |\mathbb{T}’_n|)^2)$
>     - memory: $\mathcal{O}((|Z^\text{latent}| + |Z^\text{context}| + N \cdot |\mathbb{T}’_n|)^2)$
> * Ours:
>     - time: $\mathcal{O}((|Z^\text{latent} | + | Z^\text{context}| + \sum_{n=1}^N |\mathbb{T}_n|)^2)$
>     - memory: $\mathcal{O}((|Z^\text{latent}| + |Z^\text{context}| + \sum_{n=1}^N |\mathbb{T}_n|)^2)$
>
> Where, usually, $|\mathbb{T}’_n| \gg |\mathbb{T}_n|$.
>
> ---
>
> [A]: Zhou, Dewei, et al. "MIGC++: Advanced multi-instance generation controller for image synthesis." IEEE TPAMI. 2024
>
> [B]: Zhou, Dewei, et al. "DreamRenderer: Taming multi-instance attribute control in large-scale text-to-image models." ICCV. 2025.
>
> [C]: Phung, Quynh, Songwei Ge, and Jia-Bin Huang. "Grounded text-to-image synthesis with attention refocusing." CVPR. 2024.
>
> [D]: Dahary, Omer, et al. "Be yourself: Bounded attention for multi-subject text-to-image generation." ECCV. 2024.

---

> > ### Author Rebuttal · Reviewer_K2rb · 2026-04-01
> >
> > The authors resolved my concerns.

---

> > > ### Author Response · Authors · 2026-04-07
> > >
> > > We thank the reviewer for the time provided. We’re glad that the provided clarifications adequately address the concerns and we hope they will consider reassessing their score in light of this discussion.

---

### Decision · Program_Chairs · 2026-04-30

**Decision:**

Accept (regular)

**Comment:**

This paper addresses the issue of attribute leakage in multi-instance image editing for Flow Matching models. The authors propose Instance-Disentangled Attention (IDAttn), which applies layer-wise masking to isolate regional edits in middle layers while maintaining global coherence in early and late layers.

Reviewers appreciated the introduction of a novel infographics editing benchmark and the practical efficiency of the approach through single-pass disentangled editing.

Reviewers initially raised concerns, questioning whether attribute leakage was a significant real-world problem and pointed out potential issues with evaluation heuristics and bounding box constraints. During the rebuttal phase, the authors effectively addressed these issues by providing extensive literature citations, robustness ablations on metrics, and clarifying that the method can also utilize segmentation masks. These additions resolved the concerns of most reviewers.

The rebuttal successfully clarified several points, yet Reviewer 5UCs ultimately retained their recommendation to reject the paper. The core unresolved issue for this reviewer lies in the incremental nature of the technical novelty, since the primary methodological contribution of applying attention masking to prevent feature leakage is already a well-established practice in the field of diffusion models, meaning the proposed adaptation relies heavily on an existing engineering trick rather than providing sufficient methodological depth. While the AC partially agrees with the reviewer regarding the incremental methodological innovation, we believe that the method's practical efficiency and the valuable new Infographics Editing benchmark provide sufficient utility for the community. Therefore, the paper is weakly accepted to the conference. We ask the authors to incorporate the promised changes in the camera-ready version.